# TOPS-speed complex-valued convolutional accelerator for feature extraction and inference

Yunping Bai[1], Yifu Xu [1], Shifan Chen[1], Xiaotian Zhu[2], Shuai Wang[1], Sirui Huang[1], Yuhang Song[1], Yixuan Zheng[1], Zhihui Liu[1], Sim Tan[3], Roberto Morandotti [4], Sai T. Chu [2], Brent E. Little[5], David J. Moss [6] ✉, Xingyuan Xu [1] ✉ & Kun Xu[1] ✉

Complex-valued neural networks process both amplitude and phase information, in contrast to conventional artificial neural networks, achieving additive capabilities in recognizing phase-sensitive data inherent in wave-related phenomena. The ever-increasing data capacity and network scale place substantial demands on underlying computing hardware. In parallel with the successes and extensive efforts made in electronics, optical neuromorphic hardware is promising to achieve ultra-high computing performances due to its inherent analog architecture and wide bandwidth. Here, we report a complex-valued optical convolution accelerator operating at over 2 Tera operations per second (TOPS). With appropriately designed phasors we demonstrate its performance in the recognition of synthetic aperture radar (SAR) images captured by the Sentinel-1 satellite, which are inherently complex-valued and more intricate than what optical neural networks have previously processed. Experimental tests with 500 images yield an 83.8% accuracy, close to in-silico results. This approach facilitates feature extraction of phase-sensitive information, and represents a pivotal advance in artificial intelligence towards real-time, high-dimensional data analysis of complex and dynamic environments.

Artificial neural networks (ANNs) emulate the human brain's learning processes, comprising interconnected nodes that adapt based on data. With sufficient computing power and training data, ANNs excel at tasks such as pattern recognition, speech processing, and decision-making[1–6]. While generic ANNs are real-valued, exclusively handling amplitude information, wave-related scenarios that involve both amplitude and phase information highly desire neural networks tailored for waves (i.e., complex-valued neural networks). Applications where complex-valued neural networks are heavily needed include radar technologies, which rely on understanding phase information for target detection and localization, such as analyzing ice thickness or industrial activities at sea using synthetic aperture radar (SAR) images captured by satellites[7,8]; telecommunications, where the intricate interplay of amplitude and phase defines signal characteristics; and robotics, where precise wave-based sensing enhances spatial awareness[9–14].

The complex-valued operations in neural networks are generally decomposed as real-valued multiply-and-accumulate operations, which can be achieved through reading and writing data back-and-forth between the memory and processor in von Neumann

[1]State Key Laboratory of Information Photonics and Optical Communications, Beijing University of Posts and Telecommunications, Beijing, China. [2]Department of Physics, City University of Hong Kong, Hong Kong, China. [3]School of Electronic and Information Engineering, Beihang University, Beijing, China. [4]INRS-Énergie, Matériaux et Télécommunications, Varennes, QC, Canada. [5]QXP Technology Inc., Xi'an, China. [6]Optical Sciences Centre, Swinburne University of Technology, Hawthorn, VIC, Australia. ✉e-mail: dmoss@swin.edu.au; xingyuanxu@bupt.edu.cn; xukun@bupt.edu.cn

architectures. As the data capacity (such as for massive satellite networks) and neural network scale (such as for Large Language Models) dramatically increase, the underlying computing hardware of complex-valued neural networks are expected to feature more advantages such as: (a) efficient computing architectures/interfaces compatible with waves and complex-valued data; (b) sufficiently large fan-in/out, needed for processing high-dimensional data in practical wave-related scenarios; (c) high bandwidth/throughput, for analysis of fast-varying features of waves in real-time.

In parallel with the failure of Moore's Law and thus extensive efforts made in electronics[15–18], optical neuromorphic computing hardware[19–44], offering ultra-high speeds facilitated by the >10 THz wide optical band, and minimal energy consumption down to $2.5 \times 10^{-19}$ J per operation due to their analog architecture[19,20], are promising for complex-valued neural networks. Recently, decent advances have been made with optics, such as complex-valued matrix multiplication operations using delicate photonic integrated circuits[42] or diffractive optics[22,43], and complex-valued activation functions involving 2D materials[44]. However, existing work primarily focuses on the acceleration of fully-connected networks (i.e., formed by matrix multiplication operations), and thus the data input dimension is limited by the hardware parallelism. Optical neuromorphic hardware capable of processing high-dimensional, high-speed complex-valued data streams, or fast-varying waves, have not been demonstrated yet.

Here, we report a complex-valued optical convolution accelerator (CVOCA), capable of extracting high-dimensional ultrafast hierarchical features of waves or complex-valued data streams. We propose and demonstrate methods to map complex-valued information onto analog optical physical systems in a fast manner, including complex-valued electro-optic modulation for input data and wavelength synthesizing for convolutional weights. By interleaving high-speed time-multiplexed, complex-valued input data and spectrally synthesized wavelengths from a microcomb source[45–51], we achieve a computing speed of over 2 Tera operations per second (TOPS). Further, we use this system to process SAR images captured recently by the Sentinel-1 satellite (https://sentinel.esa.int/web/sentinel/copernicus/sentinel-1), which are inherently complex-valued and intricate for inference[7–9,52–54]. Experimental tests with 500 samples yielded an accuracy of 83.8%, close to in-silico results. This universal feature extractor or data compressor, with its high-performance and efficient neuromorphic hardware tailored for intricate phase-sensitive data (or wave) processing tasks, will impact many applications from telecommunications to radar systems and satellite imaging, where it can be placed in satellites for the Copernicus program, enabling faster and better understanding of the Earth[7,8].

## Results
### Principle of operation
For complex-valued input data $\mathbf{X} = |\mathbf{X}| \cdot \exp[j \cdot \boldsymbol{\varphi}] = \mathbf{X_R} + \mathbf{X_I}$ ($\mathbf{X_R} = |\mathbf{X}| \cdot cos\boldsymbol{\varphi}$; $\mathbf{X_I} = |\mathbf{X}| \cdot sin\boldsymbol{\varphi}$) and kernel weights $\mathbf{W} = |\mathbf{W}| \cdot \exp(j \cdot \boldsymbol{\theta}) = \mathbf{W_R} + j \cdot \mathbf{W_I}$ ($\mathbf{W_R} = |\mathbf{W}| \cdot cos\boldsymbol{\theta}$; $\mathbf{W_I} = |\mathbf{W}| \cdot sin\boldsymbol{\theta}$), the convolution results can be given as $\mathbf{Y} = [|\mathbf{X}| \cdot \exp(j \cdot \boldsymbol{\varphi})]*[|\mathbf{W}| \cdot \exp(j \cdot \boldsymbol{\theta})] = (\mathbf{W_R}*\mathbf{X_R} - \mathbf{W_I}*\mathbf{X_I}) + j \cdot (\mathbf{W_R}*\mathbf{X_I} + \mathbf{W_I}*\mathbf{X_R})$, which demand fast complex-valued multiplication operations and a sliding window across the input data $\mathbf{X}$ achieved by the underlying hardware accelerator. During complex-valued convolution operations, the input data need to be multiplied by complex-valued weights in sequence (i.e., following the sliding window manner, Fig. 1), which involves simultaneous multiplications of modules (i.e., $|\mathbf{X}| \cdot |\mathbf{W}|$) and summation of phases (i.e., $\boldsymbol{\varphi} + \boldsymbol{\theta}$). The entire complex-valued convolution operation demands massive amplitude and phase operations, intrinsically requiring accurate and delicate power and phase manipulations in the optical domain. In addition, all the above operations need to be accomplished at high speeds, thereby

effectively achieving the acceleration of complex-valued convolutional calculations. As such, this complicated computing operator poses more substantial requirements on the underlying computing hardware, in contrast to the real-valued counterparts.

Here, we propose a high-performance CVOCA capable of accelerating the complex-valued convolution operations at a high-speed for inference tasks (Fig. 2). Specifically, the primary challenge to implement the CVOCA lies in mapping the complex-valued weights onto the physical optical system. To achieve this, we propose a "synthetic wavelength" method to construct complex-valued weights $\mathbf{W}$ in a stable and incoherent manner. Each synthetic wavelength (i.e., complex-valued weight) $\mathbf{W}[m]$ ($m \in [1, M]$) comprises of two micro-comb lines with the power adjusted as $\mathbf{W_R}[m]$ and $\mathbf{W_I}[m]$. We shape and interleave an on-chip microcomb as even and odd wavelength channels ($M$ wavelengths for each path) to carry the real and imaginary components of $\mathbf{W}$ (i.e., optical power equals to $\mathbf{W_R}[m]$ at $\lambda_{even}[m]$ and $\mathbf{W_I}[m]$ at $\lambda_{odd}[m]$, respectively, $m \in [1, M]$). While different wavelength channels are orthogonal (i.e., the corresponding Nyquist bandwidth, or half of adjacent wavelengths' spacings, is larger than the electrical bandwidth of input data $\mathbf{X}$), the orthogonality between the real and imaginary parts of $\mathbf{W}$ is guaranteed.

In parallel, the input data vector $\mathbf{X} = |\mathbf{X}| \cdot \exp(j \cdot \boldsymbol{\varphi}) = \mathbf{X_R} + j \cdot \mathbf{X_I}$ is time-multiplexed with a symbol duration of $T$, with its real and imaginary parts encoded onto a cosine wave with fast-varying amplitudes and phases as $\mathbf{X}[n] = |\mathbf{X}[n]| \cdot cos\{\omega_c t + \boldsymbol{\varphi}[n]\}$ ($\omega_c = 2\pi/T$ denotes carrier's angular frequency, $n$ denotes discrete temporal locations of the symbols). This cosine wave can also be given as $\mathbf{X}[n] = \mathbf{X_R}[n] \cdot cos\omega_c t + \mathbf{X_I}[n] \cdot sin\omega_c t$, denoting that the real and imaginary parts of input data $\mathbf{X}$ are carried by a pair of orthogonal bases $cos\omega_c t$ and $sin\omega_c t$, respectively.

Then, the input wave $\mathbf{X}[n]$ and its Hilbert transform $j \cdot \mathbf{X}[n]$ are modulated onto wavelengths $\lambda_{odd}$ and $\lambda_{even}$ that correspond to $\mathbf{W_R}$ and $\mathbf{W_I}$, respectively, thus yielding weighted replicas $\mathbf{W_R}[m] \cdot \mathbf{X}[n]$ and $\mathbf{W_I}[m] \cdot j \cdot \mathbf{X}[n]$, across the two sets of wavelengths.

Here we designed a complex-valued electro-optic modulator (CVEOM) to perform the electro-optic modulation of complex-valued input data and kernel weights. It has two optical input ports (for the input of $\lambda_{odd}$ and $\lambda_{even}$, respectively), two parallel sub-Mach-Zehnder modulators (for the input of $\mathbf{X}[n]$ and $j \cdot \mathbf{X}[n]$) and one optical output port. The CVEOM featured imbalanced delays between the two modulation paths to compensate for the delay difference between $\lambda_{odd}$ and $\lambda_{even}$ (induced by subsequent dispersion). We note that the CVEOM is a non-trivial device, not only for convolution accelerators demonstrated in this work but also for other neuromorphic or communications applications involving complex-valued data. Although we employed discrete fiber-based components to build the CVEOM (including two Mach-Zehnder Modulators, a 90° electrical hybrid coupler with an in-phase output $\mathbf{X}$ and a quadrature output $j \cdot \mathbf{X}$, and a tunable optical delay line to compensate for the delay differences), we note that it features similar components as classic IQ modulators such that it can be massively produced as well; nonetheless, the "wavelength synthesizing" technique of CVEOM enables manipulating the phases of optical carriers, rather than just the input signals as IQ modulators do. We note that the performance/consistency of the CVEOM can be further optimized for multi-wavelength operation, with readily available techniques such as waveguide designs[55] and feedback bias controllers[56].

Next, the weighted replicas at all wavelengths are progressively delayed via dispersion, with delay steps equal to the symbol duration $T$. As the delay difference between $\lambda_{odd}$ and $\lambda_{even}$, induced by dispersion, are compensated inside the CVEOM, the delayed replicas of $\mathbf{X}[n]$ and $j \cdot \mathbf{X}[n]$ are aligned in time, given as $\mathbf{W_R}[m] \cdot \mathbf{X}[n-m+1]$ and $j \cdot \mathbf{W_I}[m] \cdot \mathbf{X}[n-m+1]$, respectively. Finally, all weighted and delayed replicas are summed upon photodetection, yielding the output

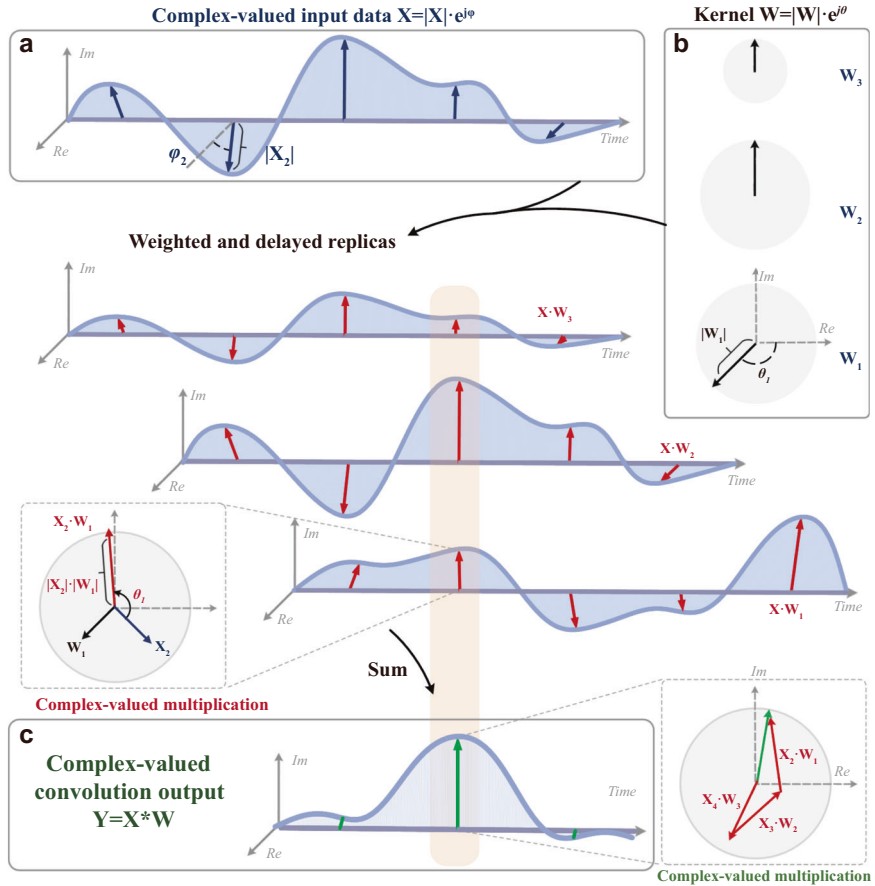

**Fig. 1 | Illustration of the complex-valued convolution process. a** The arrows represent the complex-valued input data **X** originated from wave-related signals, and the length and direction of the arrows show the amplitude and complex angle of data. **b** The complex-valued kernels with amplitude and phase. **c** The illustration shows the process of complex-valued convolution operation, wherein the outputs (green arrows) come from the complex-valued summation of multiple delayed weighted replicas.

convolution results

$$
\begin{aligned}
\mathbf{Y}[n] &= [|\mathbf{W}[m]| \exp[j \cdot \theta[m]]]^{*}[|\mathbf{X}[n]| \exp[j \cdot \varphi[n]]] \\
&= \sum_{m=1}^{M} |\mathbf{W}[m]| \cdot |\mathbf{X}[n-m+1]| \cdot \exp[j \cdot (\theta[n] + \varphi[n-m+1])] \\
&= \sum_{m=1}^{M} \mathbf{W_R}[m] \cdot \mathbf{X_R}[n-m+1] + j \cdot \mathbf{W_R}[m] \cdot \mathbf{X_I}[n-m+1] \\
&\quad + j \cdot \mathbf{W_I}[m] \cdot \mathbf{X_R}[n-m+1] - \mathbf{W_I}[m] \cdot \mathbf{X_I}[n-m+1] \\
&= \{(\mathbf{W_R}*\mathbf{X_R})[n] - (\mathbf{W_I}*\mathbf{X_I})[n]\} + j \cdot \{(\mathbf{W_R}*\mathbf{X_I})[n] + (\mathbf{W_I}*\mathbf{X_R})[n]\}
\end{aligned}
\tag{1}
$$

where the real and imaginary parts are carried on the amplitude and phase information of the output wave **Y** (i.e., the pair of orthogonal bases $\cos\omega_c t$ and $\sin\omega_c t$ after decomposition).

As such, the convolution window effectively slides across the input data at a speed equal to its baud rate $1/T$. This can be further denoted by its computing speed, given as $2 \times M/T \times 4$ operations per second (OPS), which linearly scales with the signal baud rate and number of employed wavelengths.

Notably, the CVOCA realized mapping the complex-valued weights (involving amplitude and phase) onto the synthetic optical wavelengths via the CVEOM, which can efficiently manipulate the amplitude and phase of input complex-valued data by controlling the optical power of two sets of wavelengths $\lambda_{odd}$ and $\lambda_{even}$. This approach avoids direct manipulation of optical phases, which are susceptible to external environments and remain challenging to be accurately controlled, and supports intensity detection to directly obtain the output

calculated results, thus leading to higher robustness and potentially lower cost, in contrast to coherent schemes. As a result, a convolution accelerator for complex-valued data can be achieved in this work, in a high-speed, robust, and low-cost manner.

## Complex-valued matrix convolution

In the experimental demonstration of our CVOCA, a key building block is the soliton crystal microcomb source that yielded tens of wavelength channels for the mapping of the complex-valued convolutional kernel weights **W** (Fig. 3). The employed soliton crystal microcomb, featuring a 50.2 GHz free spectral range, originated from wideband parametric oscillation in a micro-ring resonator (MRR), offering a small footprint, a large Nyquist bandwidth and a large number of wavelengths. We interleaved the microcomb as two sets of comb lines ($\lambda_{odd}$ and $\lambda_{even}$), each with $M = 9$ comb lines (18 in total) to implement a $3 \times 3$ complex-valued convolution kernel. For a proof-of-concept demonstration, we designed 4 convolution kernels based on a classic real operator **S** and its transpose $\mathbf{S^T}$ (note $\mathbf{S \cdot S^T} = 0$). Each weight matrix was flattened into a vector for comb spectra shaping via a Waveshaper.

To evaluate the performance of the CVOCA, we tested two sets of input data, including: (a) an input data matrix $\mathbf{X_a}$ constructed from the conjugates of the four kernel weight matrices $\mathbf{W^*}$ and their Hilbert transform $j \cdot \mathbf{W^*}$, such that the output feature map $\mathbf{Y_a}$ can illustrate the process of multiplication or dot product (and thus convolution) of complex-valued matrices and the capability of simultaneously processing amplitude and phase (i.e., real and imaginary) information (Fig. 3, see full results in Supplementary information); (b) a blood cell

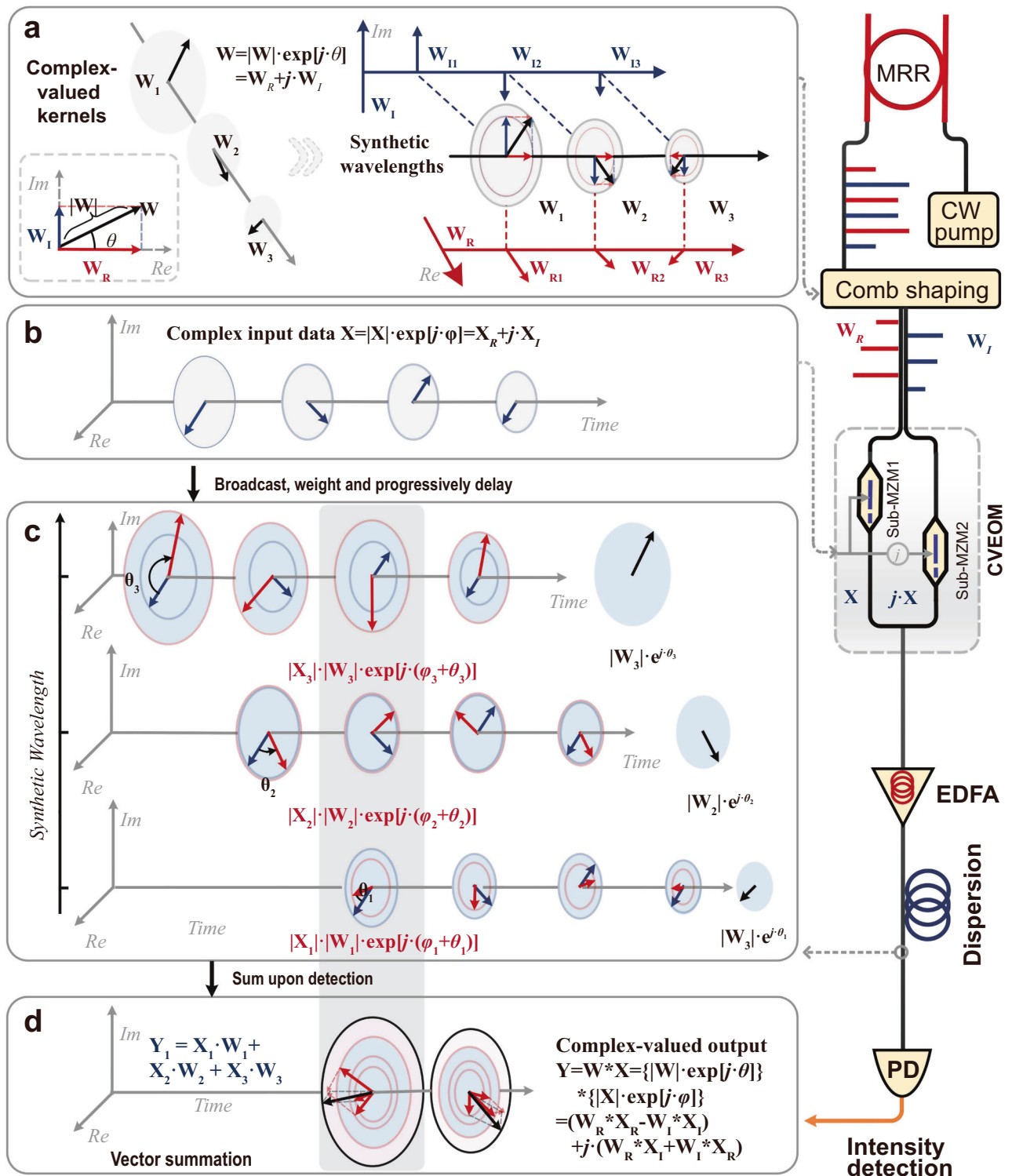

**Fig. 2 | Operation principle of the CVOCA.** Consisting of the experimental setup (right panel) and the corresponding signal flow of the complex-valued convolution process (left panel). The arrows denote phasors in the complex-valued plane. **a** The generation of synthetic wavelengths via utilizing two sets of wavelengths in the process of comb shaping. **b** The discrete complex-valued data used for modulating the synthetic wavelengths. **c** The process of complex-valued convolution calculation in the fiber transmission. **d** The output results after photoelectric detection. CW Pump: continues-wave pump laser. EDFA, erbium doped fibre amplifier. MRR, micro-ring resonator. CVEOM, complex-valued electro-optical modulator. PD, photodetection.

image $X_b$ captured via fluorescence microscopy (https://github.com/Shenggan/BCCD_Dataset), with the real and imaginary components constructed as its horizontal and vertical derivatives, to demonstrate the CVOCA's capability for practical datasets (Fig. 4, see full results in Supplementary information).

In the experiments, the input complex-valued data matrix was subsequently flattened and encoded onto a cosine wave's amplitude and phase information. Then the electrical wave was input into the CVEOM, where it and its Hilbert transform were individually modulated onto the interleaved wavelength sets $\lambda_{odd}$ and $\lambda_{even}$, yielding

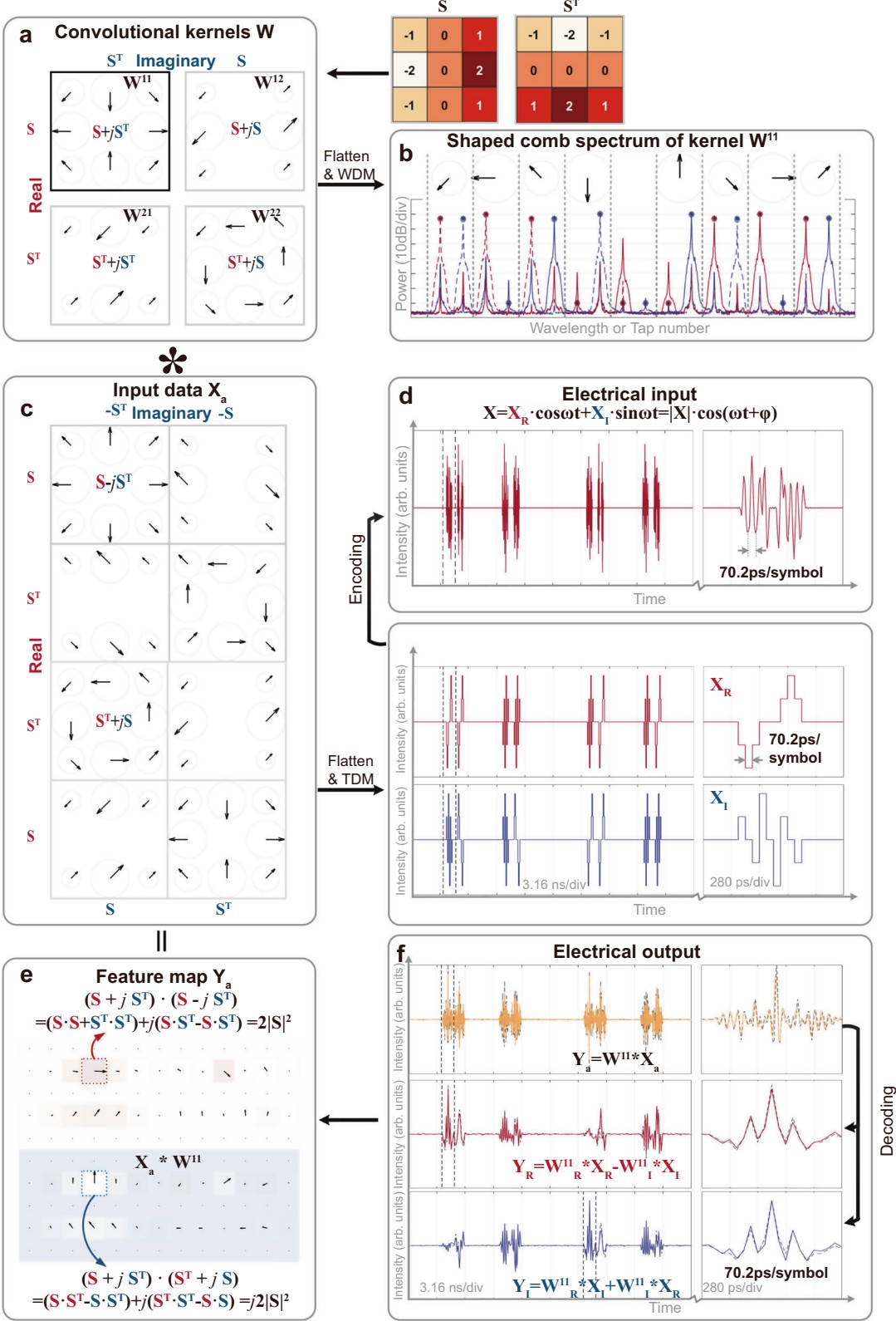

**Fig. 3 | Experimental results of CVOCA with input data $X_a$. a** Four convolution kernels $W^{11}$, $W^{12}$, $W^{21}$, and $W^{12}$ were designed based on a real kernel $S$ and its transpose $S^T$: $W^{11} = S + j \cdot S^T$, $W^{12} = S + j \cdot S$, $W^{21} = S^T + j \cdot S^T$, $W^{22} = S^T + j \cdot S$. **b** The optical spectrum shows the shaped comb for kernel $W^{11}$, with solid and dashed lines corresponding to positive and negative weights, and dots and stars denoting measured and desired weights. **c** The designed input data based on a real kernel $S$ and its transpose $S^T$. **d** The electrical waveforms generated from the input data. **e** The output feature of complex-valued convolution between the input data and kernel $W^{11}$. **f** The electrical waveforms show the convolution results, with the experimental and simulated results denoted as solid and dashed lines, respectively. The feature maps are experimental results for kernel $W^{11}$, featuring distinct locations of maximum real and imaginary elements, as highlighted in dashed boxes.

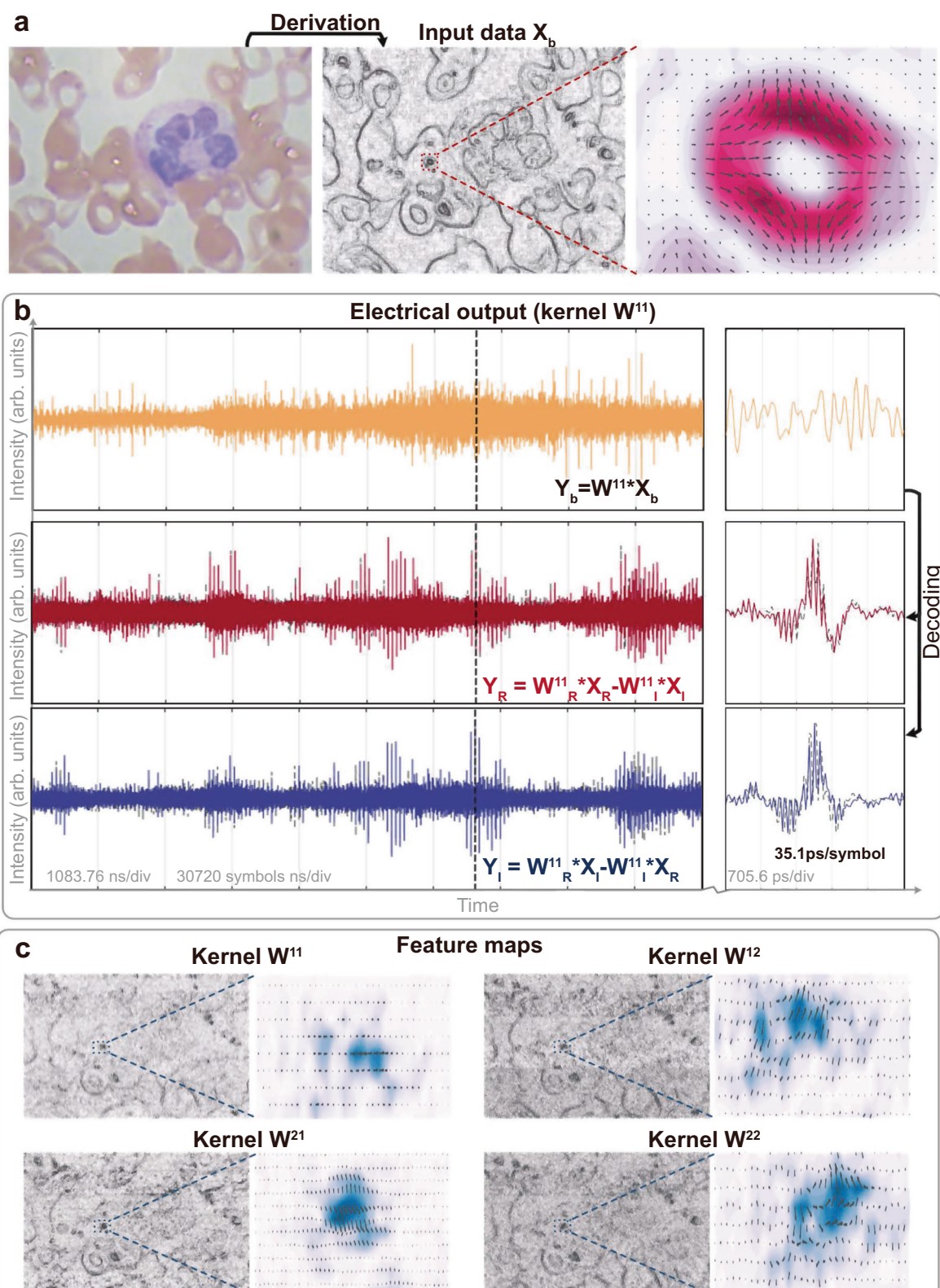

**Fig. 4 | Experimental results of CVOCA with input data $X_b$. a** The input complex-valued blood cell image is constructed from the horizontal and vertical derivatives of the original image captured via fluorescence microscopy. **b** The electrical waveforms show the convolution results between the input data and kernel $\mathbf{W^{11}}$, with the experimental and simulated results denoted as solid and dashed lines, respectively. **c** The feature maps are experimental results for the four kernels, featuring distinct features as highlighted in the zoom-in view.

weighted replicas, which were then coupled together and transmitted through a spool of dispersion compensation fiber. The delay step between adjacent wavelengths, introduced by dispersion, matched with the symbol duration, thus achieving the time-wavelength interleaving needed for convolution operations. Notably, the $\lambda_{odd}$ and $\lambda_{even}$ wavelength channels exhibited a relative delay of $T/2$ due to dispersion, which was compensated with a tunable optical delay line inside the CVEOM. Finally, the delayed replicas were summed upon photodetection to yield convolution results.

After decoding, the real and imaginary components of the output were obtained. We note that to achieve negative weights, the delayed replicas were separated into two spatial paths by a Waveshaper, according to the sign of $\mathbf{W}$, then converted to electrical signals via a balanced photodetector. Feature map matrices were obtained after resampling, where each element representing the sum of the Hadamard product between the kernel weight matrix and the input matrix's elements within the convolution window, or receptive field. As such, when the convolution window aligned with: (a) the region in $\mathbf{X}$ that equals to the kernel's conjugate, the maximized real output $|\mathbf{W}|^2$ was yielded ($2|\mathbf{S}|^2$ in Fig. 3); (b) the region in $\mathbf{X}$ that equals to the kernel's conjugated Hilbert transform, the maximized imaginary output $j\cdot|\mathbf{W}|^2$ was yielded ($j2|\mathbf{S}|^2$ in Fig. 3). This is illustrated in the feature maps $\mathbf{Y_a}$ yielded by the four kernels (Fig. 3), where distinct locations of the maximum real and imaginary elements correspond to hierarchical complex-valued features extracted by different kernels. Such characteristics of complex-valued convolution are also denoted in the processing of biological images in Fig. 4. Notably, while the strides of the convolution window were inhomogeneous due to the matrix-flattening process, they did not hinder the performance of our approach (serving as a subsampling function for pooling) and can be tailored as generic homogeneous strides when necessary.

We note that the CVOCA's performance was investigated under different system settings, involving the symbol rate $1/T$ (14.245 GBaud for $\mathbf{X_a}$, $14.245 \times 2$ GBaud for $\mathbf{X_b}$), the spacing of $\lambda_{odd}$ and $\lambda_{even}$ (200 GHz for $\mathbf{X_a}$, 100 GHz for $\mathbf{X_b}$; which indicates that the same dispersive medium can be used for both cases and thus the input data rate can be adjusted by simply reprogramming the Waveshaper and change the wavelength channels' spacing). The cosine wave's frequency $\omega_c/2\pi$ was set as 14.245 GHz for both cases. The peak computing speed of the system reaches $2 \times M/T \times 4 = 2 \times 9 \times 14.245\,\text{G} \times 4 = 1.0256$ TeraOPS for $\mathbf{X_a}$ and $2 \times 1.0256 = 2.0512$ TeraOPS for $\mathbf{X_b}$. This is the fastest complex-valued computing hardware so far.

## Complex-valued convolutional neural network

To validate the capability of the CVOCA in complex-valued feature extraction, we used it to accelerate the first convolutional layer of complex-valued convolutional neural networks (CVCNNs) for both a benchmarking handwritten digits dataset and a SAR (Synthetic Aperture Radar) imaging dataset, wherein we used two different system settings as introduced above (case $\mathbf{X_b}$ for handwritten digits, and case $\mathbf{X_a}$ for SAR images), achieving a single-kernel computing speed up to 2.0512 TeraOPS, over 3 times faster than prior photonic convolution accelerators used in inference tasks[30,33,34].

For the first handwritten digit dataset[57], the full CVCNN includes one complex-valued convolutional layer and one fully-connected layer. Each input complex-valued image with a size of $14 \times 28$ originates from folding single real-valued $28 \times 28$ images[58], wherein half data were regarded as the real parts, and the other data formed the imaginary parts (Fig. 5). The input complex-valued images are convoluted with two parallel $3 \times 3$ complex-valued convolutional kernels, and then generating two complex-valued feature patterns. Here, the CVOCA's parameters is set the same as the above test for input data $\mathbf{X_b}$, yielding the same computing speed of over 2 TOPS. The corresponding experimental results for 500 images yield a recognition accuracy of 91%, close to in-silico results.

We note that the MNIST dataset serves as a benchmark test to validate the reach of our complex-valued convolution accelerator. Due to the process of converting real-valued input data into complex-valued data (where the real and imaginary parts do not necessarily correlate with each other in practice), the recognition accuracy degraded in contrast to real-valued neural networks[58]. The method of real-to-complex conversion needs to be tailored and further optimized according to specific datasets (i.e., correlations of the raw input data) and tasks, to obtain performance improvements in contrast to real-valued neural networks.

Furthermore, we also used the CVOCA to accelerate a CVCNN specifically tailored for recognizing SAR images (https://sentinel.esa.int/web/sentinel/copernicus/sentinel-1)[7–9,52–54], wherein complex-valued convolution operations account for over 90% of the overall computing power. SAR images present a challenge as they are inherently complex-valued in their raw forms – non-trivial in contrast to benchmark datasets such as handwritten digits. This tailored utilization of the CVCNN within the SAR context underscores the critical role played by complex-valued convolution operations, further emphasizing the efficacy of our approach in addressing the computational demands of intricate tasks, particularly in SAR image recognition. The SAR images' real and imaginary components exhibit inherent correlations, jointly conveying the physical properties of the detected objects, thus CVCNNs can significantly outperform real-valued CNNs, which typically neglect the imaginary information (https://sentinel.esa.int/web/sentinel/copernicus/sentinel-1)[7–9,52–54].

The demonstrated SAR images dataset was captured by the Sentinel-1 satellite of the Copernicus program, encompassing 7 categories of Earth surfaces, including agriculture, forest, high-density urban, high-rise building, low-density urban, industry region, and water region (https://sentinel.esa.int/web/sentinel/copernicus/sentinel-1)[52–54]. Each sample includes two complex-valued $100 \times 100$ SAR images, obtained from two polarization channels of the satellite (horizontal-horizontal, or HH, and horizontal-vertical, or HV).

For both polarization channels, each input SAR image was convolved with four $3 \times 3$ complex-valued kernels, yielding four $34 \times 100$ feature maps (because of non-symmetric convolution strides, Fig. 6). Subsequent network operations were performed in silico. We note that although the kernels were sequentially implemented in the experiments, parallel operation is straightforward to implement by duplicating the current paths. The matrix-flattening methods, device parameters, and signal processing flow remained consistent with demonstrations in the above section for input data $\mathbf{X_a}$. The experimental results closely matched with in-silico results, except for that our CVOCA's input data rate was at 14.245 GBaud – over ten times' faster than its electronic counterparts (generally at ~1 GHz clock rate) and capable of processing ~13.7 million $100 \times 100$ SAR images per second.

During Post-resampling, the extracted feature maps were further processed in silico to yield recognition results (Fig. 7). We experimentally tested 500 samples (i.e., $2 \times 500$ complex-valued SAR images) and obtained a classification accuracy of 83.8%, close to the 85.4% achieved in silico. We note that the recognition accuracy (i.e., the accuracy of the demonstrated complex-valued convolution accelerator) mainly depends on the accuracy of the experimental system's time/frequency response, which was subject to factors including: weight control accuracy (subject to non-ideal wavelength-division responses of modulators, photodetectors, and amplifiers, compensated for here by the peripheral comb shaping system); the delay errors between the CVEOM's two arms (experimentally compensated for using delay lines and reduced to $ps$ level) and adjacent wavelength channels (induced by high-order dispersion, negligible in our case using a spool of dispersion compensation fiber); inter-symbol interference caused by system bandwidth limitations/nonlinearities – a common issue in optical communications that can be compensated for

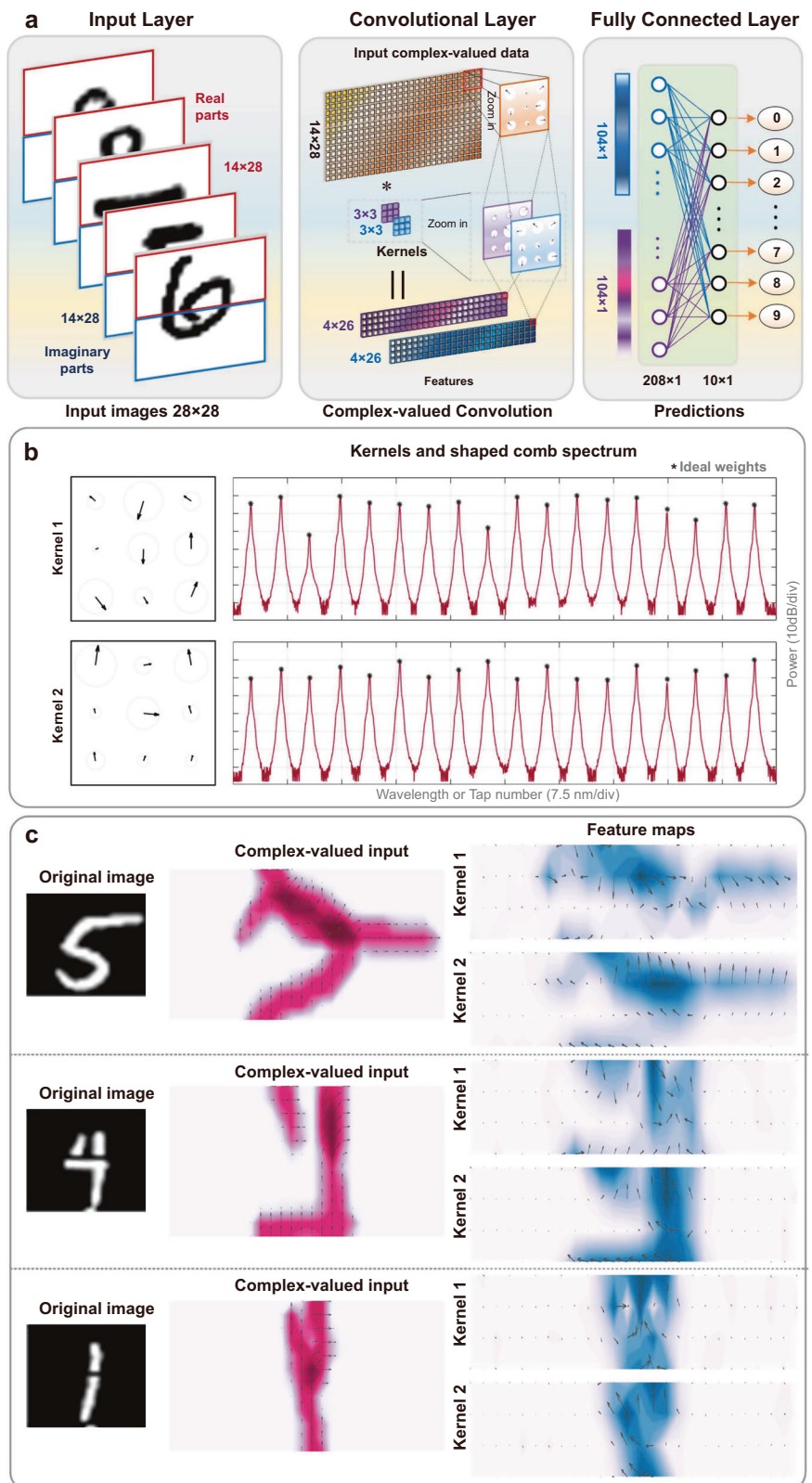

**Fig. 5 | Experimental results of handwritten digit recognition. a** The input complex-valued images originate from folding single real-valued images, where half of the data are regarded as the real parts, and the other half form the imaginary parts. The complex-valued convolutional neural network consists of one complex-valued convolutional layer and one fully-connected layer. **b** The used comb lines are shaped according to the designed kernels. **c** Two different convolutional kernels extract the features from the complex-valued images and generate feature maps, respectively.

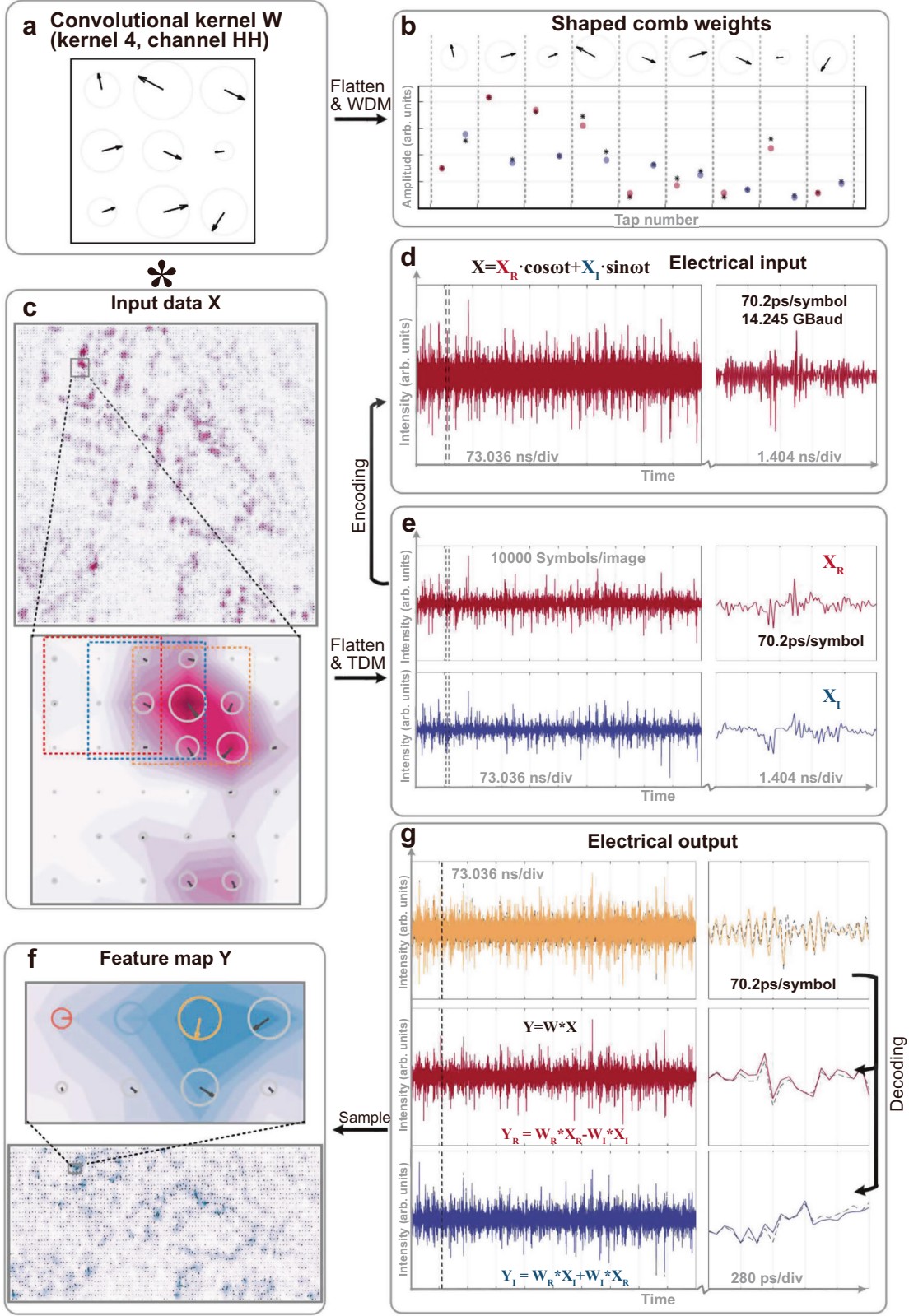

**Fig. 6 | Experimental results of SAR image recognition (kernel 4, HH channel).**
**a** The used kernel 4 in the channel HH. **b** The corresponding shaped comb weights.
**c** The input SAR image where the dashed boxes on the zoom-in input data matrix
denote the sliding convolution window and correspond to the highlighted ele-
ments in the zoom-in feature maps. **d** The input electrical waveform includes both
real parts and imaginary parts. **e** The electrical waveforms represent the real parts
and imaginary parts, respectively. **f** The output feature map. **g** The electrical output
where the experimental and simulated results are denoted as solid and dashed grey
lines, respectively.

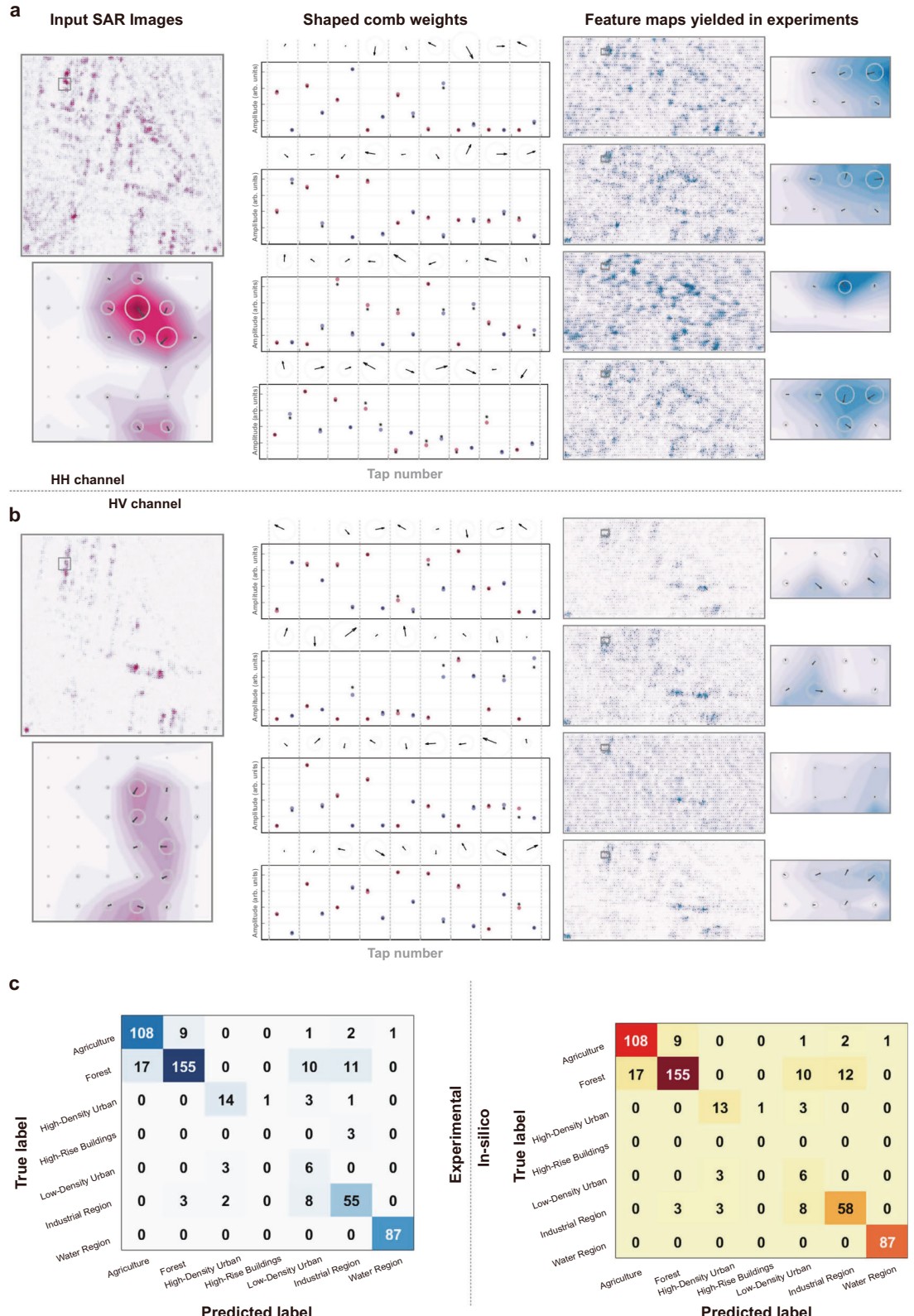

**Fig. 7 | Experimental results of SAR image recognition. a** The input images, convolution kernels, and corresponding feature maps of HH channels. **b** The input images, convolution kernels, and corresponding feature maps of HV channels.

**c** The experimental and in-silico predicted label where the confusion matrices with the darker colors indicate a higher recognition score.

via post-digital electronics. This experimental validation of CVOCA for SAR image recognition not only shows the hardware's ability in capturing intricate features of complex-valued data but also highlights its practical utility in real-world applications such as Earth observation and remote sensing.

## Discussion

In addition to its demonstrated performance, the CVOCA can be scaled to achieve significantly higher levels of parallelism and performance using readily available commercial off-the-shelf technology. By utilizing the full S, C, and L telecommunications bands (1460–1620 nm, >20 THz), we can accommodate over 200 channels (with a 100 GHz spacing). This capability can be further enhanced by leveraging polarization and spatial dimensions, enabling speeds in the PetaOPS regime. Moreover, while the comb source is already integrated, other essential components such as the optical spectral shaper, complex-valued modulator, dispersive media, demultiplexer, and photodetector can also be integrated using advanced nanofabrication techniques[59–64], enabling monolithic integration of the CVOCA with high computing speed and low SWaP (Size, Weight, and Power consumption).

In addition, while a single convolutional layer was demonstrated for the image recognition tasks in this work to validate the performance of the CVOCA, this does not indicate any limitations onto the network's scale or completeness. On one hand, the CVOCA can serve as a computing unit, or a chiplet when integrated, that can be invoked iteratively in a complicated (digital-analog hybrid) computing system, thus do not impose any limitations onto the achievable neural network scale; on the other hand, for other neuromorphic functions such as nonlinear activations[44], fully-connected layers[42] have readily been implemented using optics, which can be integrated together with the CVOCA to form a complete optical neural network.

Further, we note that this CVOCA represents major advances over the previous work[30] including:

a. Capability of extracting features of complex-valued data, or waves. This is enabled by innovative data mapping techniques, such as the CVEOM for data input and wavelength synthesizing technique for the kernel weights, enabling encoding both amplitude and phase information onto analog physical hardware in a fast manner. Followed by delicate time-wavelength interleaving of complex-valued data and weights, the complex-valued convolutional window can slide across the input data, at a speed of 28 Giga symbols per second, yielding feature maps of input waves with a computing speed of over 2 TOPS per kernel, opening up possibilities for real-time processing of waves in applications ranging from satellite imaging to telecommunications. We note that, although a complex-valued convolution is constituted by four real-valued convolutions that can be separately accelerated with the approach in ref. 30 (i.e., $\mathbf{W}^*\mathbf{X} = [\mathbf{W_R}^*\mathbf{X_R} - \mathbf{W_I}^*\mathbf{X_I}] + j\cdot[\mathbf{W_R}^*\mathbf{X_I} + \mathbf{W_I}^*\mathbf{X_R}]$), our approach that directly processes complex values is more efficient/compact. Specifically, on one hand, four separate systems are needed if using the real-valued approach[30], thus significantly increasing the overall complexity in terms of data fan-in/-out, delay error compensation, weight control, signal synchronization etc.; on the other hand, our approach is compatible with waves (i.e., $\mathbf{X}[n]|\cdot\cos\{\omega_c t + \boldsymbol{\varphi}[n]\}$), thus having the potentials of bypassing AD/sampling/demodulation processes and directly processing raw complex-valued data from communications and SAR systems, albeit requiring further investigations in terms of data encoding/decoding protocols etc.

b. Optimized performances. This work achieves a higher single-kernel computing speed for inference tasks, due to the enhanced data rate and additional parallelism brought about by complex-valued operations. In performing handwritten digits recognition, this CVOCA operates at over 2 TOPS, which is about 4 times that of

previous work[30] (5 × 5 kernel, 2 × 25 × 11.9 G = 0.595 TOPS) and even higher than other convolution accelerators[33,34].

c. More intricate inference tasks. This CVOCA was not only investigated in the CVCNN by utilizing benchmark datasets (i.e., handwritten digits) but also demonstrated in achieving extracting features of complex-valued SAR images captured by a satellite. Handwritten digits are relatively simple datasets with basic spatial structure features (e.g., the pixel value for the background in each handwritten digit image is simply zero). In contrast, real-world applications exhibit a wide variation in pixel values and possess more complex spatial structures. This complexity requires the convolutional kernel to capture features in each image more accurately for classification in the fully-connected layer. Although only the initial complex-valued convolutional layer of the CVCNN for recognizing SAR images was accelerated by utilizing the CVOCA, the complexity and difficulty are both far beyond handwritten digits recognition. Specifically, for the initial complex-valued convolutional layer the CVCNN, 500 samples with the size of 2 × 100 × 100 need to be calculated with four parallel 3 × 3 complex-valued kernels in both HH and HV channel, which is equivalent to perform the calculation of 80000 handwritten digits with the size of 28 × 28.

In conclusion, we demonstrate a complex-valued optical convolution accelerator with a computing speed exceeding 2 TOPS and use the hardware for the recognition of complex-valued synthetic aperture radar (SAR) images recently captured by the Sentinel-1 satellite. The system can process 13.7 million 100 × 100 SAR images per second, while experimental tests with 500 samples yielded an accuracy of 83.8% – close to in-silico results. This approach offers neuromorphic hardware capabilities for phase-sensitive feature extraction, with applications in telecommunications as well as in radar and satellite image processing.

## Methods

In this work, we employ a specific category of microcombs known as soliton crystals[48], which naturally form within micro-cavities featuring appropriate mode crossings, eliminating the need for intricate dynamic pumping and stabilization methods, and thus being suitable for WDM-based applications including optical neural networks. The coherent soliton crystal microcomb was generated through optical parametric oscillation within a single integrated micro-ring resonator (MRR), which was fabricated on a CMOS-compatible doped silica platform[30], featuring a high Q factor exceeding 1.5 million and a free spectral range of ~50 GHz. The pump laser at ~1570 nm was amplified to initiate parametric oscillation in the MRR, yielding over 40 channels in the telecommunications C-band (1540–1570 nm).

In the experiment, to achieve the designed kernel weights, the generated microcomb was shaped in power using two spectral shapers based on liquid crystal on silicon (Finisar WaveShaper 4000S and 16,000A). The first was used to roughly shape and interleave the microcomb lines for subsequent separate modulation, while the second achieved precise comb power shaping and negative weights together with a balanced photodetector. Specifically, the second spectral shaper precisely shaped the comb lines' power according to the absolute value of weights, then separated the comb lines into two groups according to the signs of the kernel weights. The two groups of wavelengths were directed to two separate output ports of the Waveshaper and input into a balanced photodetector (Finisar BPDV 2120 R). The balanced photodetector detected the optical power of the negative-sign and positive-sign wavelength groups, and performed differentiation of the yielded photocurrents, effectively achieving subtraction of the two wavelength groups and thus negative weights. A feedback loop was employed to improve the accuracy of comb shaping, where the error signal was generated by first measuring the

impulse response of the system and comparing it with the ideal channel weights.

A dispersive compensation module with a dispersion of 43 ps/nm was used to progressively delay the weighted replicas. Finally, according to the signs of kernel weights, the second spectral shaper routed the wavelength channels into the two input ports of a balanced photodetector (Finisar BPDV 2120 R). The output electrical temporal waveform was received and sampled by a high-speed oscilloscope (Lecroy).

## Data availability
The experimental data generated in this study are provided in the Supplementary Information.

## Code availability
The codes that support the findings of this study are available from the corresponding authors upon request.

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

## Acknowledgements

Y.B. acknowledges support by the National Natural Science Foundation of China (Grant No. 62301074), K.X. acknowledges support by the National Key R&D Program of China (Grant No. 2021YFF0901700), the National Natural Science Foundation of China (Grant No. 61821001, Grant No. 62135009), X.X. acknowledges support by the Fund of State Key Laboratory of IPOC (BUPT) (No. IPOC2023ZZ01), D.J.M. acknowledges support by the Australian Research Council (ARC) Centre of Excellence in Optical Microcombs for Breakthrough Science, COMBS (No. CE230100006). R.M. Acknowledges support from NSERC and the Canada Research Chair Program. We thank Yanni Ou, Yuyang Liu, Mingzheng Lei, Liyan Wu, and Yanlu Huang for helpful discussions.

## Author contributions

X.X. and Y.B. conceived the idea. Y.B. and Y.X. performed the experiments, with assistance from S.H., Y.S., and Y.Z.S.C. constructed the neural network model. X.Z., S.T.C., and B.E.L. designed and fabricated the integrated micro-ring resonator. S.W., Z.L., and S.T. experimentally generated the microcombs. R.M. contributed to the development of the experiment and to the data analysis. X.X., Y.B., and D.J.M. wrote the manuscript. K.X. and X.X. supervised the project.

## Competing interests

The authors declare no competing interests.
