## [Transparent Peer Review file · Nature Communications]

TOPS-speed complex-valued convolutional accelerator for feature extraction and inference

Corresponding Author: Professor xingyuan xu

Version 0:

Reviewer comments:

Reviewer #1

(Remarks to the Author)

The authors present in detail and with many interesting results a novel method for complex-valued convolutional acceleration in a photonic scheme based on MRR combs and a new type of modulation. The idea relies on a previous work for real-valued convolutional processing (ref. 30 of the manuscript) and extends the concept by splitting wavelengths and assigning real and imaginary weights to them and by introducing the CVEOM modulator for applying the input signal in an unconventional manner. Although the concept is impressive and the results support adequately the operation capabilities, I believe that the authors should address the following issues

1. The authors claim that their scheme is practical as it avoids issues related to phase instabilities. On the other hand, they double the number of required resources as they do not exploit the complex nature of light. Why a scheme where each wavelength could carry a complex weight and the data X are also applied in a complex manner through a high speed commercially available I/Q modulator is not possible? Since the wavelengths of the comb are mutually locked, phase instabilities should be minimal. Have the authors considered such a possibility?
2. The authors do not explain adequately how they imprint negative weights and separate them from the positive ones. They should be a little bit more specific probably in the supplementary material about the resources (wavelengths, photodetectors) required in order to support both positive and negative weights.
3. In CNN processing there is always the parameter of stride. Do the authors apply proper pre-processing of the data when they flatten, in order to take into account the stride parameter? How could this pre-processing increase the complexity of the process? They also write that "Notably, while the strides of the convolution window were inhomogeneous due to the matrix flattening process, they did not hinder the performance of our approach (serving as a subsampling function for pooling) and can be tailored as generic homogeneous strides when necessary." I think this is a very strong statement. They should become more specific and explain that, at least in the supplementary part.
4. The authors claim that using a CNN and a fully connected in simulations in the MNIST task, leads to an accuracy of 92% approximately. According to my experience and the literature (<https://spj.science.org/doi/full/10.34133/icomputing.0032>), a simple fully connected layer can provide almost 92% of accuracy in MNIST task and the addition of a CNN as a front-end dimensionality reduction module can further boost the performance. They should try to explain why the CNN they have considered does not contribute to the improvement of the accuracy compared to a FCL (784->10).
5. The authors do not explain how they implemented the CVEOM and its tunable optical delay. Is it a fiber-based device based on discrete components? Is it an integrated device? Moreover, it is not clear how they apply j^*X in one of the two modulators. Do they set the bias at a point which provides a phase shift of $\pi/2$? Is this easy to keep constant for all wavelengths in the comb ?
6. The authors did not compare their computation efficiency with that referred in ref. 40 where a FWM-based complex convolutional engine is presented. This is important as ref. 40 claims similar TOPs performance to that of the present work.

Reviewer #2

(Remarks to the Author)

This paper reports a complex-valued optical convolution accelerator (CVOCA). The authors utilize a microresonator frequency comb to perform computational processing of complex-valued data. The complex data is mapped onto the comb lines using electro-optic modulation in their CVOCA.

It is mentioned that a wave shaper is used to equalize the amplitude of the frequency comb lines. Is a spectrum of this available in the manuscript? The supplementary information only shows the raw generated soliton crystal state but not the spectrum after the wave shapers.

Complex-valued weights are needed for the convolution operation. The authors utilize two comb lines for this purpose. If the power in the comb lines fluctuate, for example if the microcomb state fluctuates, what would be the impact on the CVOCA operation? Or is the power maintained using the wave shaper? Supplementary information describes their selection process for the comb lines, but was there a specific reason why the two adjacent lines selected were picked? Was there a wavelength/power or other requirement?

The CVOCA is used in the first convolutional layer of CVCNNs to accelerate the operation. The work in this section of the manuscript is quite interesting. Please define SAR at the first instance it appears in the text. The accuracy of the classification operations done was good with a large sample size. Can the authors comment on how they can improve the accuracy, or what were some factors which caused the accuracy to be lower than in silico.

The authors discuss integration of the CVOCA improving the performance by polarization multiplexing. The soliton crystal is generated with a specific polarization. Please comment on how the output of the microresonator may be controlled to create the polarization states needed for polarization multiplexing, especially if integration is the final goal.

I recommend publishing after the above points are addressed.

Reviewer #3

(Remarks to the Author)

Dear editor,

In this manuscript, the authors present a complex-valued optical convolutional system capable of 2 TOPS of computational speed for data processing. The system leverages a microcomb for wavelength generation/multiplexing as well as a weighted and time-delayed signal mechanism to realize complex-valued convolution operations. Processing of SAR images and handwritten digit recognition are demonstrated with the system shown, with accuracies similar to those obtained with conventional electronic neural networks.

While the results are promising, there are two major areas that would benefit from further analysis and explanations. First of all, the motivation behind the selection of the specific method presented for handling complex-valued data should be justified. Comparisons should be made between the current implementation and a more traditional approach where the real and imaginary parts of the complex number are treated as two separate real numbers, as this more traditional approach would allow the authors to directly use their previous demonstrations [R1]. Secondly, a more detailed discussion of power use, energy efficiency metrics, and system implementation/scaling cost should be provided to place the presented results in better context of the state-of-the-art currently available. I have provided detailed comments regarding these two issues and some other relevant points below:

1. The statement in the abstract regarding processing of complex valued data is subjective and can potentially be interpreted as an exaggeration.

a. While there are some challenges, most of them can be dealt with separating the real and imaginary parts of the complex numbers and processing them simultaneously/separately as two independent real numbers. This naturally requires additional memory; however, that typically does not present insurmountable challenges in traditional electronic neural networks. Existing literature includes many such examples. I suggest the authors reword their phrasing in the abstract.

b. I also believe the introduction would benefit from a discussion of existing methods of processing complex valued data in fully in-silico networks. This would also help place the presented work in better context of existing electronic solutions (not just optical/physical-domain solutions), and better highlight the claimed advantages.

2. The presented idea is primarily based on delayed and weighted signal replicas measured through incoherent detection, as was shown previously by the authors. This allows signals encoded onto multiple different wavelengths to “interfere” with one another, therefore creating the ability to extract spatial information from the provided data. While the idea is now well established, there are still some drawbacks that need to be discussed in detail.

a. In delay-weight-sum type of networks, aspects regarding the symbol overlap and delay timing impose limitations on the system’s information processing bandwidth. Currently, is this bandwidth primarily limited by data modulation and detection speeds? If so, are there any other fundamental limitations on the optical system’s capability of processing information at higher speeds?

b. The high implementation cost (high speed modulators, waveshapers for realizing convolution kernels, requirements of balanced photodetection for negative weights, multiple amplifiers necessary etc.) can present critical challenges for the adoption of such systems. These additional requirements also indicate a substantial power budget for the system presented. Can the authors discuss these implementation aspects in more detail? Other than the modulators and the detectors, are there any other components that can be replaced with on-chip equivalents, to reduce power and footprint requirements?

3. The demonstrated system deals with complex valued data as two separate data streams being fed into a pair of MZIs modulating spectrally adjacent wavelengths of light, that are then subsequently delayed according to the symbol rate. While this implementation provides a way to process data coming in as real-and-imaginary pairs, it is similar to how one would deal with two separate streams of data, or even a single stream of data that is separated onto two distinct channels. In that case, would it be possible to use the authors' demonstration from their earlier results, with a fully real-valued approach to mimic processing of complex valued data? This is an important point that needs to be clarified, especially to properly distinguish this

4. Similar to my comment above, a complex valued multiply-and-accumulate (MAC) operation is not fundamentally different from a set of real valued MAC operations. There are several fundamental and implementation-related perspectives that need to be addressed regarding this aspect:

a. Firstly, this fact is already explicitly stated by the equation in line 76 of the paper, where the real and imaginary parts of the result (which are separately real-valued) are simply a collection of the MAC operation results between the real and imaginary parts (which are also separately real-valued) of W and X . From this perspective, by separating the real and imaginary parts, one can execute complex MAC using only real-valued operations. This indicates that while hardware demands may increase slightly due to the extra multiplications necessary, the actual system (electrical or optical) is not fundamentally different from one that performs real-valued MACs. This is an important aspect that needs more detailed and clear explanations in the manuscript.

b. Secondly, the use of a pair of wavelengths (named odd and even) indicates that the operation being performed is quite similar to (likely the same as) separating the complex data into two streams of real numbers, and processing them in the same way as demonstrated before in [R1]. This should be investigated in detail, and the comparison of these two approaches should be provided in the manuscript. Currently, the reader is left questioning the efficacy of the presented approach.

[R1] Xu, X. et al. 11 TOPS photonic convolutional accelerator for optical neural networks. *Nature* 589, 44–51 (2021)

5. Is it possible to implement a similar processing architecture using IQ modulation? In the current implementation, the delayed overlaps of different wavelengths is what enables feature extraction. At the same time, the system requires synthesis, weighing, and dispersion control of twice as many wavelengths. Naturally, it would be advantageous to explore the possibility of using a single wavelength for each data, but incorporate other orthogonal modalities such as IQ modulation.

6. An important of optical networks like the one presented here is their inherent capability to handle theoretically unlimited precision, whereas their electronic counterparts are typically limited to double-precision arithmetic. Similarly, while increased precision incurs computational overhead in electronics, the detector in an optical system practically does not care about the input precision, given that it receives sufficient power. Can this advantage potentially be leveraged to perform operations that require computationally prohibitive precision levels in electronics, in an optical manner through this weighed-and-delayed processing method? Even if that is not currently demonstrated, the authors should discuss the possibility of other potential tasks that require more intensive electronic computations, but can be performed in fewer optical layers using a system such as the one demonstrated here.

7. From a more general standpoint, many authors in modern physical (or physically-inspired) machine learning literature discuss the benefits and drawbacks of two information processing regimes: In the first regime, the physical networks constructed (like the one shown in this paper) exactly replicate the operations that would otherwise be done in electronic circuits through completely artificial neural networks. In the second regime, the physical nature (including complex system dynamics, memory, nonlinearity etc.) of the constructed system directly performs various inference tasks, with potential added pre- or post-processing through electronic layers. While there may not be clear winner between these two regimes, it is quite important to discuss the capabilities of the presented system in the context of these two approaches. Especially since the presented demonstration mimics an operation that can already be performed by existing electronics, it is critical to provide broader application perspectives in the second regime aforementioned. For instance, can a system like the one demonstrated here be leveraged for other operations that are currently not possible through conventional electronics? In the presented work, are there any system dynamics (optical nonlinearity, memory, inter-modal coupling etc.) that could be taken advantage of in the future?

8. The hand-written digit recognition has become a staple in demonstration of many physical machine learning models, as also shown in Fig 5. Looking at the details shown here, it appears that the original data starts as a real-valued image, and is then converted into a complex-valued image through a slicing operation. This method of converting real data into complex data is quite interesting, but I am not aware of any physical or mathematical motivation for this choice of operation. It would make a lot more sense (and be commensurate with well-known approaches in image processing) to perform FFT on these images, and retrieve the amplitudes and phases of the transforms to use as complex-valued data in such a problem. In that case, it is also possible that the underlying feature maps obtained carry physically relevant information regarding both the geometrical structures in the original image and their specific locations. In fact, it is well-established that the phase of the transform carries more information that is relevant to human perception than the amplitude. Since it is done electronically, is there a reason that the authors did not opt for a more conventional real-to-complex data conversion method here? Is it possible that the authors present capabilities of the network (at least in simulation), using instead a spectral transform such as FFT?

9. Fig 7's comparison to in-silico metrics for the SAR images is mainly focused on the prediction accuracy of 83.8% to demonstrated electronic accuracy of 85.4%. However, even before constructing the optical system, one expects these metrics to be similar as the optical system exactly mimics the mathematical operations performed by its electronic counterpart. One also does not expect the optical system to significantly outperform its electronic counterpart on accuracy alone, due to the same reason above, since the fundamental computations are identical. These points highlight that a more detailed comparison including other aspects is necessary. Energy efficiency is presented as one of the most important advantages of the demonstrated system. As such, it is critical to include quantitative comparisons regarding the amount of energy per operation (or per image, per bit, etc.) used in the demonstrated system, and place it in context of existing state-of-the-art electronic or optical systems in the literature.

10. On a related point, while the discussion of power is currently missing from the manuscript; and the power characteristics of the microcomb are also not reported. Can the authors please report the total optical input power and/or power per comb line? Understanding these metrics is important for evaluating the energy efficiency and scalability of the system. Then, the authors should discuss if any steps can be taken to further reduce the optical power necessary, relevant optical losses in the system, and other factors contributing to the overall power budget for the image processing capabilities presented. Finally, at least the obtained SNR at the detectors should be reported, in order to fully convey the technical details necessary.

11. Even though the authors discuss the potential for scaling up to Peta-OPS performance, aren't there potential bottlenecks regarding inter-modal crosstalk? Also, couldn't more closely spaced wavelengths be used for stronger parallelization? Does the reduced spectral spacing between channels pose any restrictions for this purpose?

12. The numbering of references in the main text should be revised to reflect their order of appearance.

Version 1:

Reviewer comments:

Reviewer #1

(Remarks to the Author)

The authors have in depth discussed and answered in all my comments. The paper now is by far improved and convincing in terms of the principle and the results. I recommend the acceptance of the paper in Nature Communications.

Reviewer #2

(Remarks to the Author)

The authors have addressed my comments. I recommend publishing the manuscript.

Reviewer #3

(Remarks to the Author)

I thank the authors for the changes they implemented in the manuscript, as they have addressed all of my comments (and the other reviewers' comments) in detail. I appreciate the discussion of IQ modulation possibility, the clarification of real-to-complex data conversion details, and potential pathways for monolithic integration. It is now more clear what advantages the provided system can have by processing complex-valued data as is, using complex-valued kernels. I believe the article and the supporting material are now appropriate for publication at Nature Communications.

Detailed response to comments from reviewers

Manuscript ID: Nature Communications manuscript NCOMMS-24-38220

We thank the reviewers for their thoughtful suggestions and positive comments that we believe have significantly helped us improve the manuscript. We address their comments in detail here and have marked the changes in **red** in the revised manuscript accordingly.

We've summarized the changes that have been made to the manuscript and supplementary materials, as the following table shows:

Major concerns	Corresponding comments	Revisions
More information /clarifications needed	Rev. 1, point 2, 3, 4, 5; Rev. 2, point 1, 2; Rev. 3, point 9, 10;	Implementation of negative weights added in Methods; Discussions on the convolution stride added in the SM; Discussions on the MNIST dataset added in the Manuscript and SM; Implementation of CVEOM added in the Manuscript; Shaped microcomb spectra added in the SM; Discussion regarding microcombs added in the SM; Discussion regarding potential integration and power budget added in the SM;
More discussions needed	Rev. 1, point 1, 6; Rev. 2, point 3,4; Rev. 3, point 1b, 2a, 2b, 3, 4a, 4b, 5, 6, 7, 8, 11;	Discussion regarding IQ modulation added in the SM; Discussion regarding ref. [40] added in the SM; Discussion regarding recognition accuracy added in the SM; Discussion regarding microcomb polarization added in the SM; Discussion regarding complex NNs added in the Manuscript; Discussion regarding advances over prior arts [30] added in the Manuscript; Discussion regarding precision added in the SM;

		Discussion regarding physical computing systems added in the SM; Discussion regarding real-to-complex conversion added in the SM; A new figure illustrating potential schemes added in the SM; Discussions regarding channel spacing added in the SM;
Refinement of claims	Rev. 3, point 1a, 12;	Claims in Abstract adjusted; Reference list updated;

Before responding to the reviewers' comments in detail, we recap the breakthroughs of this paper. We report:

- first experimental demonstration of complex-valued optical neuromorphic convolution hardware accelerator;
- fastest complex-valued photonic computing hardware at a computing speed exceeding 2 TOPS;
- capable of processing 13.7 million 100-by-100 SAR images per second —over ten times higher than electronics (at ~1GHz clock rate);
- first demonstration of highly complex and intricate SAR imaging datasets captured by the Sentinel-1 satellite.

Reviewer #1 (Remarks to the Author):

The authors present in detail and with many interesting results a novel method for complex-valued convolutional acceleration in a photonic scheme based on MRR combs and a new type of modulation. The idea relies on a previous work for real-valued convolutional processing (ref. 30 of the manuscript) and extends the concept by splitting wavelengths and assigning real and imaginary weights to them and by introducing the CVEOM modulator for applying the input signal in an unconventional manner. Although the concept is impressive and the results support adequately the operation capabilities, I believe that the authors should address the following issues.

We appreciate the reviewer's very positive comments.

1. The authors claim that their scheme is practical as it avoids issues related to phase instabilities. On the other hand, they double the number of required resources as they do not exploit the complex nature of light. Why a scheme where each wavelength could carry a complex weight and the data X are also applied in a complex manner through a high speed commercially available I/Q modulator is not possible? Since the wavelengths of the comb are mutually locked, phase instabilities should be minimal. Have the authors considered such a possibility?

We appreciate the reviewer's insightful comments. We agree that I/Q modulation would use less wavelength channels. We've added discussions in the Supplementary Materials to introduce potential coherent architectures for complex-valued convolution, and the involved tradeoffs in contrast to the employed incoherent approach.

Supplementary Materials

Discussions

“Coherent architecture. We note that the complex-valued convolution can also be implemented via coherent architectures, which uses a single wavelength for each element of the complex-valued weight \mathbf{W} , as illustrated in Fig. S15. The microcomb is first split into two paths, one path goes with an IQ modulator to load the complex-valued data's real and imaginary components, the other path goes through a Waveshaper such that the carriers' amplitudes and phases can be simultaneously manipulated to implement the complex-valued weight \mathbf{W} . A coherent receiver is necessary to obtain the complex-valued convolution results.

The main features/differences of the I/Q modulation scheme, in contrasted to the incoherent approach demonstrated in this work, include: a) baseband modulation format that directly loads the complex-valued data \mathbf{X} ; b) coherent detection that requires an additional optical path to provide the LO (i.e. microcombs with complex-valued weights \mathbf{W}).

We note that, while the coherent architecture uses a single wavelength for each element of the complex-valued weight \mathbf{W} , it's subject to the nontrivial challenge — LO phase instability and noises, as commonly faced by coherent optical communications. This arose

due to the fluctuations/noises of relative optical phases between the signal and LO paths, and requires either optical phase locked loops or/and post-DSP (for real-time phase retrieval and compensation) — both significantly increase the complexity and cost of the system, and cost nontrivial additional computing power for error/gradient calculation, phase retrieval and compensation etc.

The proposed incoherent approach, specifically the “synthetic wavelength” method, can address the issues brought about by the coherent architecture. Our incoherent approach constructs complex-valued weights \mathbf{W} in a stable and incoherent manner, where the complex-valued data input and results output are independent from optical phases of the carriers (i.e., shaped microcombs), offering significantly enhanced stability, weight accuracy and robustness without additional phase locked loops or DSP. We also note that the scheme proposed in this work can also directly process waves with fast varying amplitudes and phases $X[n] = |X[n]| \cdot \cos\{\omega_c t + \phi[n]\}$, rather than just complex-valued vectors $X_R[n]$ and $X_I[n]$ supported by the baseband IQ modulation format.”

Figure S15 | Example of coherent computing architecture for complex-valued convolution.

2. The authors do not explain adequately how they imprint negative weights and separate them from the positive ones. They should be a little bit more specific probably in the supplementary material about the resources (wavelengths, photodetectors) required in order to support both positive and negative weights.

We appreciate the reviewer’s comments and have added more details in the Methods section.

Main Manuscript

Methods

“In the experiment, to achieve the designed kernel weights, the generated microcomb was shaped in power using two spectral shapers based on liquid crystal on silicon (Finisar WaveShaper 4000S and 16000A). The first was used to roughly shape and interleave the microcomb lines for subsequent separate modulation, while the second achieved precise comb power shaping and negative weights together with a balanced photodetector. Specifically, the second spectral shaper precisely shaped the comb lines’ power according to the absolute value of weights, then separated the comb lines into two groups according to the signs of kernel weights. The two groups of wavelengths were directed to two separate output ports of the Waveshaper and input into a balanced photodetector (Finisar BPDV 2120R). The balanced photodetector detected the optical power of the negative-sign and positive-sign wavelength groups, and performed differentiation of the yielded photocurrents, effectively achieving subtraction of the two wavelength groups and thus negative weights.”

3. In CNN processing there is always the parameter of stride. Do the authors apply proper pre-processing of the data when they flatten, in order to take into account the stride parameter? How could this pre-processing increase the complexity of the process? They also write that “Notably, while the strides of the convolution window were inhomogeneous due to the matrix flattening process, they did not hinder the performance of our approach (serving as a subsampling function for pooling) and can be tailored as generic homogeneous strides when necessary.” I think this is a very strong statement. They should become more specific and explain that, at least in the supplementary part.

We agree with and appreciate the reviewer’s insightful comments. We’ve added discussions in the Supplementary Materials to clarify this point.

Supplementary Materials

Discussions

“**Convolution strides.** In this work, the convolution accelerator we designed fundamentally operates on vectors; hence, for two-dimensional image processing applications, the input data must be flattened into vectors before processing. We utilize inhomogeneous strides, where the horizontal stride within the receptive field is set to 1, ensuring that all horizontal features from the original data are extracted; the vertical stride is equal to the height of the convolution kernel, reducing the overlap when reading the input data and partially achieving the function of pooling. Similar schemes can be referred to in the relevant sections of [30] and [s19].

Specifically, in the preprocessing stage, based on the heterogeneous strides of the receptive field, we first horizontally partition the original input matrix into multiple sub-matrices, each with a height equal to the height of the convolution kernel. We then flatten each sub-matrix into a vector from top to bottom and left to right and finally concatenate these vectors end-to-end to form a complete picture corresponding vector, ensuring that the movement of the receptive field corresponds precisely with the order of data reading.

We note that the two-dimensional data that needs to be processed during CNN processing is typically stored in corresponding storage devices. Thus, the required preprocessing merely involves reading the information from the two-dimensional data in a specific order without adding extra repetitive reads or increasing the complexity of the process. While the heterogeneous strides do not limit the performance of the convolution accelerator, as evidenced by the high recognition success rate of the CNN in our fully digital predictions, homogeneous strides can be achieved by adjusting how data is read during preprocessing or by increasing the number of accelerator spatial paths [30].”

References

[30] Xu, X. et al. 11 TOPS photonic convolutional accelerator for optical neural networks. *Nature* 589, 44–51 (2021)

[s19] Krizhevsky A, Sutskever I, Hinton G E. Imagenet classification with deep convolutional neural networks. *Advances in neural information processing systems* 25, 1097-1105 (2012).

4. The authors claim that using a CNN and a fully connected in simulations in the MNIST task, leads to an accuracy of 92% approximately. According to my experience and the literature (<https://spj.science.org/doi/full/10.34133/icomputing.0032>), a simple fully connected layer can provide almost 92% of accuracy in MNIST task and the addition of a CNN as a front-end dimensionality reduction module can further boost the performance. They should try to explain why the CNN they have considered does not contribute to the improvement of the accuracy compared to a FCL (784->10).

We appreciate the reviewer’s comments. We note that the advantages/performances of our approach are demonstrated mainly by the SAR image recognition task where the input data are inherently complex-valued. The MNIST dataset serves as a benchmark test to validate the reach of our approach, during which the input data are real-valued and need to be converted into complex-valued data—this process introduces degradation in the recognition accuracy [60]. As such, although the parametric complexity of the neural network can be reduced by the convolutional layer, the recognition accuracy did not outperform real-valued neural networks such as the FCL (784->10). Nonetheless, we note that the recognition accuracy can be further improved by optimizing the real-to-complex conversion process and hyperparameters such as the learning rate, activation function, loss function etc.—deferring to our future work.

We’ve made revisions as follows to clarify this point.

Main Manuscript

Complex-Valued Convolutional Neural Network

“We note that the MNIST dataset serves as a benchmark test to validate the reach of our complex-valued convolution accelerator. Due to the process of converting real-valued input data into complex-valued data (where the real and imaginary parts do not necessarily correlate with each other in practice), the recognition accuracy degraded in contrast to real-

valued neural networks [60]. The method of real-to-complex conversion needs to be tailored and further optimized according to specific datasets (i.e., correlations of the raw input data) and tasks, to obtain performance improvements in contrast to real-valued neural networks.”

Supplementary Materials

Network training

“For the MINIST handwritten digit recognition task, data points in the top half of each image are assigned to the real part and points in the bottom half are assigned to the imaginary part. The training set consists of 60,000 images, and the testing set contains 10,000 images. The network includes a convolutional layer (2 kernels), a ReLU activation function, and a fully connected layer. The convolutional kernel size is 3×3 , and the fully connected layer comprises 208 nodes. The convolution layer has a vertical stride of 3. We trained the network using Cross-Entropy loss for 20 epochs, employing the backpropagation algorithm with Stochastic Gradient Descent. The momentum was set to 0.9, the batch size to 64, and the learning rate to 0.01. The entire network performs 32,032 operations per image, with the convolutional layer accounting for 14,976 operations, which is 45% of the total operations needed. The in-silico recognition accuracy of the experimentally test 500 dataset is 92.8%. We note that, although the recognition accuracy did not show significant improvement, in contrast to a network with a single fully connected layer (10 neurons, each with 784 synaptic weights, $784 \times 10 = 7840$ real-valued synaptic weights in total) [s1], the parametric complexity can be significantly reduced ($208 \times 10 = 2080$ complex-valued synaptic weights in the fully connected layer, $3 \times 3 \times 2 = 18$ complex-valued synaptic weights in the convolutional layer).”

5. The authors do not explain how they implemented the CVEOM and its tunable optical delay. Is it a fiber-based device based on discrete components? Is it an integrated device? Moreover, it is not clear how they apply $j \cdot X$ in one of the two modulators. Do they set the bias at a point which provides a phase shift of $\pi/2$? Is this easy to keep constant for all wavelengths in the comb?

We appreciate the reviewer's comments. As a proof of concept, the CVEOM employed in the experiment is a fiber-based device, assembled from discrete components including two Mach-Zehnder Modulators (MZMs), a 90° electrical hybrid coupler with an in-phase output X and a quadrature output $j \cdot X$, and a tunable optical delay line to compensate for the delay differences between λ_{odd} and λ_{even} (induced by subsequent dispersion). Both two modulators were biased at quadrature to achieve standard double-sideband modulation, with electrical inputs from the outputs of the 90° electrical hybrid coupler to load X and $j \cdot X$, respectively.

We note that, the bias of the modulators across all wavelengths may vary due to wavelength-dependent characteristics of the modulators, or drift due to temperature variations. However, on one hand, this didn't introduce significant distortions/instabilities as verified by the reach of experimental demonstrations; on the other hand, the modulators can be readily tailored in architecture [56] to obtain optimized performance for multi-

wavelength signals, and the bias drift can be overcome via external feedback control circuits [57].

[56] Zhang, C., et al. Ultra-broadband MMI power splitter from 1.26 to 1.67 μm with photonic bound states in the continuum. *Optics Communications* 562, 130525 (2024).

[57] Kim M., Yu B., and Choi W., A Mach-Zehnder modulator bias controller based on OMA and average power monitoring. *IEEE Photonics Technology Letters* 29, 2043-2046 (2017).

We've made revisions as follows to clarify this point.

Main Manuscript

Principle of Operation

“We note that the CVEOM is a non-trivial device, not only for convolution accelerators demonstrated in this work, but also for other neuromorphic or communications applications involving complex-valued data. **Although we employed discrete fiber-based components to build the CVEOM (including two Mach-Zehnder Modulators, a 90° electrical hybrid coupler with an in-phase output X and a quadrature output $j \cdot X$, and a tunable optical delay line to compensate for the delay differences),** we note that it features similar components as classic IQ modulators such that it can be integrated and massively produced as well; nonetheless, the “wavelength synthesizing” technique of CVEOM enables manipulating the phases of optical carriers, rather than just the input signals as IQ modulators do. **We note that the performance/consistency of the CVEOM can be further optimized for multi-wavelength operation, with readily available techniques such as waveguide designs [56 and feedback bias controllers [57].”**

6. The authors did not compare their computation efficiency with that referred in ref. 40 where a FWM-based complex convolutional engine is presented. This is important as ref. 40 claims similar TOPs performance to that of the present work.

We appreciate the reviewer's comments. We've added relevant discussions of [40] in the Supplementary Materials. We note that we didn't thoroughly discuss [40] as it didn't fully achieve complex-valued convolution operations, although it's a decent work first exploring FWM for convolution operations.

The experimental architecture in ref. 40 uses photodetectors (PD) for optoelectronic detection, which can only receive real values (i.e., power of complex-valued optical fields). As mentioned in section 3.A of ref. 40, “We extract the information on the light by photodetector (PD), so the results are proportional to the square of the magnitude of light.” Therefore, only real-valued convolution results can be received when using direct detection, indicating that its computation engine can only output real-valued convolution results. Additionally, in the experiments of [40], only real-valued (rather than complex-valued) data inputs and weights were demonstrated. On one hand, the computational architecture performs convolution of real-valued data, as it used purely intensity modulation (MZM2). On the other hand, the authors noted “adjust the WSS to make the relative phase of each carrier wave in one comb to be 0”, indicating that the phases are not adjusted to achieve

complex values for complex-valued computation but rather to ensure “the convolution results detected by the PD will have obvious characteristics of constructive interference”. Therefore, in [40], “complex-valued” refers to the intermediate four-wave mixing process, rather than the convolution operation.

In terms of the computing performance, as mentioned in the Discussion section of [40], “The experiment results in this paper are all in region V” meaning that only the Region V among the five spectral regions actually performed the required convolution operation. Consequently, the operations in the other regions do not contribute to the computation efficiency of convolution operations. According to Section S6 of the supplemental materials in [40], the number of parallel operations in a single bit time for the experimental results in Section 3.A is 5. Additionally, since the experimental results in Section 3.B focus on image processing, only C3 represents a valid convolution result. Therefore, the number of parallel operations in a single bit time for Section 3.B is also 5. As a result, when considering only the convolution results of this experiment, the computational throughput of the demonstrated experiments is $5 \times 25 \text{ G} = 0.125$ tera operations per second (TOPS) for the experiment in Section 3.A and $5 \times 30 \text{ G} = 0.15$ TOPS in Section 3.B. Besides, it is noted that the input data are the OOK signal, the computational throughput of the demonstrated experiments should be expressed as 0.125 Terabits/s for the convolution result D of input A and weight B (as shown in Equation 11 in [40]).

We’ve added discussions accordingly as follows:

Supplementary Materials

Potentials of performance scaling

“Scaling. . . . Here we also highlight possibilities of using other approaches to demonstrate complex-valued convolutions, such as using optical four-wave mixing [40], which achieved a computing speed of 0.15 TOPS (up to $111 \times 30 \text{ G} = 3.33$ TOPS when taking into consideration of all involved computing regions). We note that this scheme cannot support fully functional complex-valued convolutions in its current form, and further investigations such as complex-valued data input and output are necessary.”

Reviewer #2 (Remarks to the Author):

This paper reports a complex-valued optical convolution accelerator (CVOCA). The authors utilize a microresonator frequency comb to perform computational processing of complex-valued data. The complex data is mapped onto the comb lines using electro-optic modulation in their CVOCA.

1. It is mentioned that a wave shaper is used to equalize the amplitude of the frequency comb lines. Is a spectrum of this available in the manuscript? The supplementary information only shows the raw generated soliton crystal state but not the spectrum after the wave shapers.

We appreciate the reviewer's thoughtful comments.

In the experiments, we utilized a wave shaper to directly control and manipulate the amplitude of the frequency comb lines according to the desired values of weights. As such, the shaped comb spectra for specific kernels were given in the corresponding demonstrated tasks. Specifically, in the proof-of-concept demonstration, the shaped spectrum of kernel W^{11} was shown in the top of Fig. 3 in the main manuscript, wherein the unit of the ordinate in the shaped spectrum is dBm. The shaped spectra of the other kernels (W^{12} , W^{21} , W^{22}) in the proof-of-concept demonstration were given in Fig. S3, wherein the units of the shaped spectra (i. e., the shaped comb weights) are transferred into mW, in order to compare with weight values that are linearly represented. As for the MINIST handwritten digit recognition task, the shaped comb spectra of both kernels with the unit of dBm were given in Fig. 5 in the main manuscript, wherein the real and imaginary parts were merged within one spectrum measured before the CVEOM. Similarly, the measured shaped comb spectra in the SAR image recognition were shown in the main manuscript in Fig. 6-7.

Besides, we've included the experimentally yielded additional shaped comb spectra of the SAR image recognition task in the Supplementary Materials, as follows.

Supplementary Materials

Additional results

“Figure S13 show the experimentally yielded additional shaped comb spectra of the SAR image recognition task.”

Figure S13 | The experimentally yielded additional shaped comb spectra in the SAR image recognition task

2. *Complex-valued weights are needed for the convolution operation. The authors utilize two comb lines for this purpose. If the power in the comb lines fluctuate, for example if the microcomb state fluctuates, what would be the impact on the CVOCA operation? Or is the power maintained using the wave shaper? Supplementary information describes their selection process for the comb lines, but was there a specific reason why the two adjacent lines selected were picked? Was there a wavelength/power or other requirement?*

We appreciate the reviewer's comments. The generated Kerr microcomb operates in a stable soliton crystal oscillation state, and that the power of the comb remains constant over a certain period. In our previous work, we have measured the microcomb power stability over 66 hours, with the optical spectrum captured every 15 minutes. The extracted relative standard deviation was -14 dB over 66 hours, indicating that the microcomb source's stability can well support our convolution accelerator.

Regarding the choice of the two adjacent comb lines, this selection was made as an optimal balance between the number of comb lines used in parallel and the speed of signal processing. The free spectral range (FSR) of the integrated micro-ring resonator (MRR) fundamentally determines the computing speed of the optical accelerator due to the constraints imposed by the available optical bandwidth. In our setup, the FSR of the integrated MRR is approximately 50.2 GHz, and it generates a soliton crystal with a wavelength spacing of ~ 0.4 nm. As long as the optical comb fully utilizes the available bandwidth and the modulation bandwidth (~ 25.1 GHz in our case) matches the FSR (~ 50.2 GHz), thus resulting in a Nyquist bandwidth of ~ 25.1 GHz), the computing speed does not significantly vary with the number of comb lines or the FSR.

Furthermore, the FSR also influences the length of the delay lines and can cause power fading due to fiber dispersion effects. To address these challenges, we propose an optical

interleaving technique, as shown in Figure S1 of the supplementary material. By selecting half of the microcomb lines within the chosen optical bands as synaptic weights—by using one of the two adjacent lines—the wavelength spacing between adjacent channels in the real or imaginary optical sub-bands is equal to four times the wavelength spacing of the generated soliton crystal microcomb. The proposed wavelength interleaving technique significantly enhances the optical bandwidth of each wavelength channel, reduces the dispersive delay induced by optical fibers, and eliminates the power fading caused by optical dispersion.

We've added discussions accordingly as follows:

Supplementary Materials

Details of experiments

“The generated soliton crystal microcomb with a pump wavelength at 1570.62 nm offers over 90 channels within ~40 nm. The optical spectrum of experimentally generated optical frequency comb is given in Fig. S1(a). **The generated Kerr microcomb operates in a stable soliton crystal oscillation state, and that the power of the comb remains constant over a certain period. In our previous work, we have measured the microcomb power stability over 66 hours, with the optical spectrum captured every 15 minutes. The extracted relative standard deviation was -14 dB over 66 hours, indicating that the microcomb source's stability can well support our convolution accelerator.**

As is shown in Fig. S1(f), the operation of selecting, shaping and de-multiplexing the microcomb lines for implementing mapping the real and imaginary parts of multiple spatially parallel complex-valued convolutional kernels are actually performed by a 1×4 waveshaper (Waveshaper 4000A), where different selected microcomb sub-bands for mapping one set of synaptic weights can be output at multiple output ports. **As a result, the wavelength spacing between the adjacent wavelength channels inside real-part or imaginary part optical sub-bands is equal to the four times of wavelength spacing of the generated soliton crystal microcomb (see Fig. S2(d) and S2(e)). At the same sequence position, the wavelength differences between two optical sub-bands are always the double of wavelength spacing of the microcomb. The novel method to select the microcomb lines for mapping the weights makes the available optical bandwidth inside the single wavelength channel significantly improved without sacrificing any microcomb line and computing parallelism. More importantly, the needed dispersive delay induced by the transmitted optical fibre obtains considerable reduction, thereby leading to effectively weakening effects of the power fading arising from the fibre dispersion.”**

3. The CVOCA is used in the first convolutional layer of CVCNNs to accelerate the operation. The work in this section of the manuscript is quite interesting. Please define SAR at the first instance it appears in the text. The accuracy of the classification operations done was good with a large sample size. Can the authors comment on how they can improve the accuracy, or what were some factors which caused the accuracy to be lower than in silico.

We appreciate the reviewer's positive and insightful comments. We've made revisions accordingly to the manuscript as follows:

Main manuscript

Introduction

“Applications where complex-valued neural networks are heavily needed include radar technologies, which rely on understanding phase information for target detection and localization, such as analyzing ice thickness or industrial activities at sea using **synthetic aperture radar (SAR)** images captured by satellites [7, 8]; telecommunications, where the intricate interplay of amplitude and phase defines signal characteristics; and robotics, where precise wave-based sensing enhances spatial awareness [9-14].”

Complex-Valued Convolutional Neural Network

“Post-resampling, the extracted features maps were further processed in silico to yield recognition results (Fig. 7). We experimentally tested 500 samples (i.e., 2×500 complex-valued SAR images) and obtained a classification accuracy of 83.8%, close to the 85.4% achieved in silico. **We note that the recognition accuracy (i.e., the accuracy of the demonstrated complex-valued convolution accelerator) mainly depends on the accuracy of experimental system's time/frequency response, which were subject to factors including: weight control accuracy (subject to non-ideal wavelength-division responses of modulators, photodetectors and amplifiers, compensated for here by the peripheral comb shaping system); the delay errors between the CVEOM's two arms (experimentally compensated for using delay lines and reduced to ps level) and adjacent wavelength channels (induced by high-order dispersion, negligible in our case using a spool of dispersion compensation fiber); inter-symbol interference caused by system bandwidth limitations/nonlinearities — a common issue in optical communications that can be compensated for via post digital electronics.**”

4. The authors discuss integration of the CVOCA improving the performance by polarization multiplexing. The soliton crystal is generated with a specific polarization. Please comment on how the output of the microresonator may be controlled to create the polarization states needed for polarization multiplexing, especially if integration is the final goal.

We appreciate the reviewer's insightful comments. We've made revisions to the manuscript accordingly as follows:

Supplementary Materials

Potentials of performance scaling

“**Scaling.** ... Each channel is further split by a 1×2 optical coupler and mapped to the real and imaginary weight components through the dual-polarization multiplexing, wherein the real and imaginary weights can be encoded onto the same wavelength sets via dual-

polarization modulation. This can be achieved by splitting the microcomb (linearly polarized) in power into two paths (both with the same wavelengths), and separately shaped (according to desired weights \mathbf{W}_R and \mathbf{W}_I) and modulated (with inputs of \mathbf{X} and its Hilbert transform $j\cdot\mathbf{X}$, respectively), then one arm's polarization state is rotated to the other orthogonal polarization axis (i.e., from TE to TM) and combined together with the other arm (i.e., TE). Such architectures, using additional polarization rotators and combiners to support the polarization division multiplexing, can be readily achieved in integrated forms as demonstrated in [s3].”

I recommend publishing after the above points are addressed.

We greatly appreciate the reviewer's insightful comments and support to this work.

Reviewer #3 (Remarks to the Author):

In this manuscript, the authors present a complex-valued optical convolutional system capable of 2 TOPS of computational speed for data processing. The system leverages a microcomb for wavelength generation/multiplexing as well as a weighted and time-delayed signal mechanism to realize complex-valued convolution operations. Processing of SAR images and handwritten digit recognition are demonstrated with the system shown, with accuracies similar to those obtained with conventional electronic neural networks.

First of all, the motivation behind the selection of the specific method presented for handling complex-valued data should be justified. Comparisons should be made between the current implementation and a more traditional approach where the real and imaginary parts of the complex number are treated as two separate real numbers, as this more traditional approach would allow the authors to directly use their previous demonstrations [R1]. Secondly, a more detailed discussion of power use, energy efficiency metrics, and system implementation/scaling cost should be provided to place the presented results in better context of the state-of-the-art currently available. I have provided detailed comments regarding these two issues and some other relevant points below:

We appreciate the reviewer's constructive comments, and have made corresponding revisions in detail as follows.

1. The statement in the abstract regarding processing of complex valued data is subjective and can potentially be interpreted as an exaggeration.

a. While there are some challenges, most of them can be dealt with separating the real and imaginary parts of the complex numbers and processing them simultaneously/separately as two independent real numbers. This naturally requires additional memory; however, that typically does not present insurmountable challenges in traditional electronic neural networks. Existing literature includes many such examples. I suggest the authors reword their phrasing in the abstract.

We appreciate and agree with the reviewer's comments. We have reworded the abstract accordingly as follows:

Main Manuscript

Abstract

“Complex-valued neural networks process both amplitude and phase information, in contrast to conventional artificial neural networks, achieving new capabilities in recognizing phase-sensitive data inherent in wave-related phenomena. **The ever-increasing data capacity and network scale place substantial demands on underlying computing hardware. In parallel with the successes and extensive efforts made in electronics, optical neuromorphic hardware is promising to achieve ultra-high computing performances due to its inherent analog architecture and wide bandwidth. Here, we report ...**”

b. I also believe the introduction would benefit from a discussion of existing methods of processing complex valued data in fully in-silico networks. This would also help place the presented work in better context of existing electronic solutions (not just optical/physical-domain solutions), and better highlight the claimed advantages.

We appreciate and agree with the reviewer's comments. We have included discussions as follows:

Main Manuscript

Introduction

“The complex-valued operations in neural networks are generally decomposed as real-valued multiply-and-accumulate operations, which can be achieved through reading and writing data back-and-forth between the memory and processor in von Neumann architectures. As the data capacity (such as for massive satellite networks) and neural network scale (such as for Large Language Models) dramatically increase, the underlying computing hardware of complex-valued neural networks are expected to feature more advantages such as: a) efficient computing architectures/interfaces compatible with waves and complex-valued data; b) sufficiently large fan-in/out, needed for processing high-dimensional data in practical wave-related scenarios; c) high bandwidth/throughput, for analysis of fast-varying features of waves in real time.”

2. The presented idea is primarily based on delayed and weighted signal replicas measured through incoherent detection, as was shown previously by the authors. This allows signals encoded onto multiple different wavelengths to “interfere” with one another, therefore creating the ability to extract spatial information from the provided data. While the idea is now well established, there are still some drawbacks that need to be discussed in detail.

a. In delay-weight-sum type of networks, aspects regarding the symbol overlap and delay timing impose limitations on the system's information processing bandwidth. Currently, is this bandwidth primarily limited by data modulation and detection speeds? If so, are there any other fundamental limitations on the optical system's capability of processing information at higher speeds?

We appreciate the reviewer's comments. We have included discussions as follows:

Supplementary Materials

Potentials of performance scaling

“**Signal bandwidth.** We note that the potential analog bandwidth of input signal is subject to: a) the bandwidth of modulators and photodetectors, which can be readily achieved up to over 260 GHz [61-62, s2]; b) the Nyquist bandwidth, or half of the microcomb's free spectral range/spacing (50 GHz for a 100GHz spaced comb source).”

b. The high implementation cost (high speed modulators, waveshapers for realizing convolution kernels, requirements of balanced photodetection for negative weights, multiple amplifiers necessary etc.) can present critical challenges for the adoption of such systems. These additional requirements also indicate a substantial power budget for the system presented. Can the authors discuss these implementation aspects in more detail? Other than the modulators and the detectors, are there any other components that can be replaced with on-chip equivalents, to reduce power and footprint requirements?

We appreciate the reviewer's comments and have added further discussions accordingly.

Supplementary Materials

Potentials of performance scaling

“Monolithic Integration. Although discrete components, other than the microcomb source, were used in the proof-of-concept demonstration, all components comprising of the CVOCA can be readily integrated. The microcomb itself is an integrated circuit that arises from a CMOS-compatible platform [51]. Integrated electro-optic interfaces, including modulators and photodetectors, readily support data bandwidths over 260 GHz [61-62, s2] and dual polarization modulation [s3]. The rest components of the CVOCA, including the optical spectral shaper, dispersive media, and de-multiplexer, have all been achieved based on integrated platforms [s4-s7, 63-66]. With all components integrated, the power consumption of the CVOCA mainly comes from the light source (can reach as low as 98 mW [s8]); other active devices, including modulators (thin film Lithium Niobate [s9]), photodetectors (InP [s10]), and phase shifters (thin film Lithium Niobate [s11] or doped SOI [s12]) in the optical spectral shaper, only need bias voltages and consume negligible power.”

3. The demonstrated system deals with complex valued data as two separate data streams being fed into a pair of MZIs modulating spectrally adjacent wavelengths of light, that are then subsequently delayed according to the symbol rate. While this implementation provides a way to process data coming in as real-and-imaginary pairs, it is similar to how one would deal with two separate streams of data, or even a single stream of data that is separated onto two distinct channels. In that case, would it be possible to use the authors' demonstration from their earlier results, with a fully real-valued approach to mimic processing of complex valued data? This is an important point that needs to be clarified, especially to properly distinguish this

We appreciate and agree the reviewer's comments. We have added further discussions accordingly.

Main Manuscript

Discussions

“Further, we note that, this CVOCA is the first demonstrated complex-valued optical convolutional hardware, representing major advances over the previous work [30]

including: a) Capability of directly extracting features from complex-valued data or waves... We note that, although a complex-valued convolution constitutes of four real-valued convolutions that can be separately accelerated with the approach in [30] (i.e., $\mathbf{W}*\mathbf{X} = [\mathbf{W}_R*\mathbf{X}_R - \mathbf{W}_I*\mathbf{X}_I] + j\cdot[\mathbf{W}_R*\mathbf{X}_I + \mathbf{W}_I*\mathbf{X}_R]$), our approach that directly processes complex values is more efficient/compact. Specifically, on one hand, four separate systems are needed if using the real-valued approach [30], thus significantly increasing the overall complexity in terms of data fan-in/-out, delay error compensation, weight control, signal synchronization etc.; on the other hand, our approach are compatible with waves (i.e., $|\mathbf{X}[n]| \cdot \cos\{\omega_c t + \phi[n]\}$), thus having the potentials of bypassing AD/sampling/demodulation processes and directly processing raw complex-valued data from communications and SAR systems, albeit requiring further investigations in terms of data encoding/decoding protocols etc.”

4. *Similar to my comment above, a complex valued multiply-and-accumulate (MAC) operation is not fundamentally different from a set of real valued MAC operations. There are several fundamental and implementation-related perspectives that need to be addressed regarding this aspect:*

a. Firstly, this fact is already explicitly stated by the equation in line 76 of the paper, where the real and imaginary parts of the result (which are separately real-valued) are simply a collection of the MAC operation results between the real and imaginary parts (which are also separately real-valued) of W and X . From this perspective, by separating the real and imaginary parts, one can execute complex MAC using only real-valued operations. This indicates that while hardware demands may increase slightly due to the extra multiplications necessary, the actual system (electrical or optical) is not fundamentally different from one that performs real-valued MACs. This is an important aspect that needs more detailed and clear explanations in the manuscript.

b. Secondly, the use of a pair of wavelengths (named odd and even) indicates that the operation being performed is quite similar to (likely the same as) separating the complex data into two streams of real numbers, and processing them in the same way as demonstrated before in [R1]. This should be investigated in detail, and the comparison of these two approaches should be provided in the manuscript. Currently, the reader is left questioning the efficacy of the presented approach.

[R1] Xu, X. et al. 11 TOPS photonic convolutional accelerator for optical neural networks. *Nature* 589, 44–51 (2021)

We appreciate and agree with the reviewer’s comments. We’ve included discussions in the main manuscript to compare with the previous paper [R1] and highlight the advances made in this work. Please refer to our response to the above comment.

5. *Is it possible to implement a similar processing architecture using IQ modulation? In the current implementation, the delayed overlaps of different wavelengths is what enables feature extraction. At the same time, the system requires synthesis, weighing, and dispersion control of twice as many wavelengths. Naturally, it would be advantageous to*

explore the possibility of using a single wavelength for each data, but incorporate other orthogonal modalities such as IQ modulation.

We appreciate and agree with the reviewer's insightful comments. We have provided a detailed discussion on the IQ modulator scheme. Please refer to our response made to *Comment 1, Reviewer 1*.

6. An important of optical networks like the one presented here is their inherent capability to handle theoretically unlimited precision, whereas their electronic counterparts are typically limited to double-precision arithmetic. Similarly, while increased precision incurs computational overhead in electronics, the detector in an optical system practically does not care about the input precision, given that it receives sufficient power. Can this advantage potentially be leveraged to perform operations that require computationally prohibitive precision levels in electronics, in an optical manner through this weighed-and-delayed processing method? Even if that is not currently demonstrated, the authors should discuss the possibility of other potential tasks that require more intensive electronic computations, but can be performed in fewer optical layers using a system such as the one demonstrated here.

We greatly appreciate the reviewer's insightful comments that point out the advantages and future research directions of optical computing hardware. We've added further discusses according as follows.

Supplementary Materials

Potentials of performance scaling

“Precision. Further, we note that the input data's precision of analog optical computing hardware can be potentially much larger than their digital electronic counterparts. In contrast to the bit resolution of digital electronics that is determined by the architecture/memory width and scales with the electrical signal-to-noise ratio (*ESNR*) at one bit per 6dB of *ESNR*, optical computing hardware can potentially process data with much higher precision/bit-resolution, since: a) optical signals are less susceptible to electromagnetic interference compared to electronic signals. This can lead to clearer signal transmission and higher fidelity in data representation; b) optical computing typically generates less heat than electronic circuits, allowing for more efficient operation and the possibility of more complex systems without thermal throttling; c) optical systems can exploit principles of quantum mechanics, such as superposition and entanglement, to perform complex computations that traditional electronic systems cannot easily achieve. These advantages position optical computing hardware, including our CVOCA or more complicated optical computing hardware such as Ising machines [s18], as a promising candidate for directly processing real-world analog information (such as waves) without losing precisions due to the sampling process of digital electronics.”

7. From a more general standpoint, many authors in modern physical (or physically-inspired) machine learning literature discuss the benefits and drawbacks of two information processing regimes: In the first regime, the physical networks constructed (like

the one shown in this paper) exactly replicate the operations that would otherwise be done in electronic circuits through completely artificial neural networks. In the second regime, the physical nature (including complex system dynamics, memory, nonlinearity etc.) of the constructed system directly performs various inference tasks, with potential added pre- or post-processing through electronic layers. While there may not be clear winner between these two regimes, it is quite important to discuss the capabilities of the presented system in the context of these two approaches. Especially since the presented demonstration mimics an operation that can already be performed by existing electronics, it is critical to provide broader application perspectives in the second regime aforementioned. For instance, can a system like the one demonstrated here be leveraged for other operations that are currently not possible through conventional electronics? In the presented work, are there any system dynamics (optical nonlinearity, memory, inter-modal coupling etc.) that could be taken advantage of in the future?

We greatly appreciate the reviewer's insightful comments regarding the two information processing regimes in physical machine learning, and appreciate the suggestion to discuss our system's capabilities in the context of both regimes. We've added discussions accordingly as follows.

Supplementary Materials

Potentials of performance scaling

“Other potentials. In parallel with accelerating classic operations widely achieved in digital electronics (such as the convolutions or matrix multiplication), optical computing systems' inherent physical natures (such as complex nonlinear dynamics) can be further explored to achieve dramatically increased computing performances (trillions of physical parameters can be involved within computations in several nanoseconds), albeit with tradeoffs in terms of compatibility (such as network fan-in/-out, data formats, and universality) with existing digital electronics. We note that, while the demonstrated CVOCA performs linear operations (formed by multiplication and accumulation), it involves much more complicated nonlinear dynamics that can be further investigated and potentially harnessed to achieve dramatically different computing regimes. For example, the used microcombs arose from parametric oscillation inside a micro-ring resonator that has: a long photon life time (Q factor > 1.2 million), supporting data storage and accumulation; high nonlinearity that supports high-dimension nonlinear mapping and interference via four-wave mixing; tailored dispersion and mode-crossing that enables manipulation of the signal's linear transmission process and wavelength-dependent characteristics. Moreover, the generated soliton crystal state itself represents a mathematical solution of the Lugiato-Lefever equation, indicating that complicated physical systems can be used in turn to dramatically accelerate computing operations—if appropriate data mapping/encoding can be addressed.”

8. The hand-written digit recognition has become a staple in demonstration of many physical machine learning models, as also shown in Fig 5. Looking at the details shown here, it appears that the original data starts as a real-valued image, and is then converted into a complex-valued image through a slicing operation. This method of converting real

data into complex data is quite interesting, but I am not aware of any physical or mathematical motivation for this choice of operation. It would make a lot more sense (and be commensurate with well-known approaches in image processing) to perform FFT on these images, and retrieve the amplitudes and phases of the transforms to use as complex-valued data in such a problem. In that case, it is also possible that the underlying feature maps obtained carry physically relevant information regarding both the geometrical structures in the original image and their specific locations. In fact, it is well-established that the phase of the transform carries more information that is relevant to human perception than the amplitude. Since it is done electronically, is there a reason that the authors did not opt for a more conventional real-to-complex data conversion method here? Is it possible that the authors present capabilities of the network (at least in simulation), using instead a spectral transform such as FFT?

We appreciate the reviewer’s insightful comments. We note that the MNIST task served as a benchmark to verify the performance of the CVOCA in convolving complex-valued data, and we used the method in [60] to convert the real-valued images into complex-valued ones. However, as acknowledged in [60], such method does not guarantee a better performance than real-valued operators and needs further optimization to demonstrate its advantage/efficacy. Please refer to Reviewer#1, Comment 2.

In terms of FFT, we have actually simulated its performance, yet found this approach led to worse performance than the presented method [60], as shown in the following figure. We assume that this is because the MNIST dataset are more widely separated in the ‘spatial’ hyperspace (i.e., its raw form) than in the ‘spectral’ hyperspace (after FFT), which can be further investigated with deterministic decision boundaries using supporting vector machines. We note that, for datasets that are inherently more separable in the ‘spectral’ hyperspace, such as voices/speeches, the FFT based method should outperform the slicing approach.

Figure. R3-1 the comparison of different data-conversion methods

We've added discussions accordingly to clarify this point.

Supplementary Materials

Network training

“For the MNIST handwritten digit recognition task, data points in the top half of each image are assigned to the real part and points in the bottom half are assigned to the imaginary part. We note that here the MNIST task served as a benchmark to verify the performance of the CVOCA in convolving complex-valued data, and the real-to-complex conversion method [60] does not guarantee a better performance than real-valued operators. Further optimization of this method or alternative Fourier transform-based methods are necessary to reveal complex-valued operators' advantages for real-valued data.”

9. Fig 7's comparison to in-silico metrics for the SAR images is mainly focused on the prediction accuracy of 83.8% to demonstrated electronic accuracy of 85.4%. However, even before constructing the optical system, one expects these metrics to be similar as the optical system exactly mimics the mathematical operations performed by its electronic counterpart. One also does not expect the optical system to significantly outperform its electronic counterpart on accuracy alone, due to the same reason above, since the fundamental computations are identical. These points highlight that a more detailed comparison including other aspects is necessary. Energy efficiency is presented as one of the most important advantages of the demonstrated system. As such, it is critical to include quantitative comparisons regarding the amount of energy per operation (or per image, per bit, etc.) used in the demonstrated system, and place it in context of existing state-of-the-art electronic or optical systems in the literature.

We appreciate the reviewer's insightful comments and have added quantitative comparisons regarding the amount of energy efficiency as follows.

Supplementary Materials

Potentials of performance scaling

“**Scaling.** Although the demonstrated CVOCA readily achieve high performances among complex-valued optical hardware accelerators, its parallelism and thus computing performance can be further boosted using photonic multiplexing methods and state-of-art techniques. Specifically, as shown in Fig.S14, the number of wavelength channels can be significantly increased by using broader bandwidths of microcombs, for example, over 200 wavelength channels (at a 100 GHz spacing) can be obtained when exploring the S, C and L bands (~20-THz). Two such MRRs with 50-GHz spacing difference in center pump wavelengths are interleaved and combined through a 2×1 optical coupler to produce a light source with 400 wavelengths (at a 50 GHz spacing), then split into four parallel channels via a 1×4 optical coupler. Each channel is further split by a 1×2 optical coupler and mapped to the real and imaginary weight components through the dual-polarization multiplexing, wherein the real and imaginary weights can be encoded onto the same wavelength sets via dual-polarization modulation. This can be achieved by splitting the microcomb (linearly

polarized) in power into two paths (both with the same wavelengths), and separately shaped (according to desired weights \mathbf{W}_R and \mathbf{W}_I) and modulated (with inputs of \mathbf{X} and its Hilbert transform $j \cdot \mathbf{X}$, respectively), then one arm's polarization state is rotated to the other orthogonal polarization axis (i.e., from TE to TM) and combined together with the other arm (i.e., TE). Such architectures, using additional polarization rotators and combiners to support the polarization division multiplexing, can be readily achieved in integrated forms as demonstrated in [s3]. After spectral shaping and demultiplexing, 10 spatial parallel channels are generated, with each channel supporting four 3×3 complex-valued convolutional kernels. As such, four parallel channels include 40 spatial parallel channels in total, thereby supporting 40 3×3 kernels. For the single spatial parallel channel, the Nyquist bandwidth of input signal is over 100 GHz (can reach 500 GHz in theory) and enough for an input data rate of 100 GBaud. Therefore, the computing speed would be $100\text{G} \times (2 \times 4 \times 9 + 2) = 7.4$ Tops per kernel, and thus $7.4 \text{ Tops} \times 4 \times 10 \times 4 = 1.184$ Peta-OPs in the entire scaled CVOCA.

Energy Efficiency. It is a challenge to directly reflect the ultimate potential of our scheme due to the fact that the CVOCA in this work was validated with discrete devices. Therefore, we have evaluated the energy efficiency of a fully integrated CVOCA, which has the same architecture as the scaled CVOCA. As shown in Fig.S14, the scaled scheme requires about 47 erbium-doped amplifiers based on integrated circuits, with each amplifier's power consumption not exceeding 200 mW [63]. As such, the energy per operation of the scaled CVOCA can be roughly given as $(98 \times 2 + 47 \times 200)\text{mW} / 1.184 \text{ POPs} = 0.008 \text{ pJ/operation}$ —exceeding electronics (0.5pJ/operation [s15]). Table S1 compares the power consumption of this work with existing state-of-the-art electronic or optical systems. Such performances will reach the same level of (if not exceed) state-of-art electronics [s13-s15], capable of serving as an efficient computing unit of an electro-optic hybrid computing hardware, which leverages the broad bandwidths of optics and the flexibility of electronics, ultimately achieving unparalleled performances for artificial intelligence applications.”

Table.S1 The energy efficiency of the potential scaled CVOCA compared with state-of-the-art electronic or optical systems.

Reference	Power Consumption (pJ/operation)
Nature (2021) [30]	0.8
Nature (2021) [22]	2.5
NVIDIA A100 [s16]	0.641
Google TPU-v4 [s17]	0.699
This work (the potential scaled CVOCA)	0.008

Designed Peta-OP/s optical complex-valued convolution accelerator

Figure S14 | The schematic diagram of designed scaled optical complex-valued convolution accelerator by fully using the multiple freedom degrees of light.

10. On a related point, while the discussion of power is currently missing from the manuscript; and the power characteristics of the microcomb are also not reported. Can the authors please report the total optical input power and/or power per comb line? Understanding these metrics is important for evaluating the energy efficiency and scalability of the system. Then, the authors should discuss if any steps can be taken to further reduce the optical power necessary, relevant optical losses in the system, and other factors contributing to the overall power budget for the image processing capabilities presented. Finally, at least the obtained SNR at the detectors should be reported, in order to fully convey the technical details necessary.

We appreciate and agree with the reviewer's insightful comments. We have provided a detailed discussion on the energy efficiency in our response to above comment, and added some necessary technical details as follows.

Supplementary Materials

Details of experiments

“As for the optical power of the microcomb and the optical losses encountered within the system, we use the optical input with the power of ~ 30.5 dBm into the micro-ring to generate the microcomb in this work, and the total optical output power of the 18 comb lines used in the CVEOM scheme is ~ 20 dBm. The measured optical spectrum of the microcomb after 20dB attenuation is shown in Figure S1a, and the microcomb generation system is shown in Figure S1b. We used an erbium-doped amplifiers (EDFA) in the CVEOM scheme, which have a total power consumption of ~ 200 mW, to compensate for the optical losses of the comb shaping and modulators so that the total optical input power of the photodetectors was at ~ 7 dBm, corresponding to a root-mean-square voltage of ~ 100 mV_{rms} for the received electronic signal (with a responsivity of ~ 0.4 A/W and 50Ω impedance).

The electrical signal-to-noise ratio (ESNR) of our experimental system was mainly subject to our external equipment rather than the CVOCA itself: the arbitrary waveform generator (Keysight 8196, Effective Number of Bits = 5.4) has an effective ESNR of $5.4 \times 6 = 32.4$ dB for the computing system; the oscilloscope (Lecory 830, vertical noise floor = 2.90 mV_{rms}) has an effective ESNR of $20 \cdot \log_{10}(100/2.90) = 30.75$ dB for the computing system. Due to limited experimental capabilities for accurate ESNR characterization, we used experimentally measured waveforms to roughly estimate the ESNR of the computing system as 36.57 dB (with < 10 dB ESNR enhancement using averaging), close to the upper limit of the external equipment, thus indicating that the CVOCA would have a similar, if not higher, ESNR.”

11. Even though the authors discuss the potential for scaling up to Peta-OPS performance, aren't there potential bottlenecks regarding inter-modal crosstalk? Also, couldn't more closely spaced wavelengths be used for stronger parallelization? Does the reduced spectral spacing between channels pose any restrictions for this purpose?

We appreciate the reviewer's comments. As mentioned by our response made to your Comment 2(a) (Reviewer 3, Comment 2(a)), the potential analog bandwidth of input signal is subject to the Nyquist bandwidth, or half of the microcomb's free spectral range/spacing (50 GHz for a 100GHz spaced comb source). While more closely spaced wavelengths be used for stronger parallelization, the reduced spectral spacing between channels will indeed pose restrictions on the Nyquist bandwidth of input signal. This issue can be resolved through the wavelength interleaving method introduced in Fig. S1. Specifically, all used wavelengths with more closely spaced can be divided into 9 groups in sequence, the single parallel computing channel can be designed by selecting one wavelength at the same position from every group to make up the weights of one 3×3 kernel. In this way, multiple parallel computing channels with significantly improved wavelength space and Nyquist bandwidth can be easily obtained, with the more closely spaced wavelengths utilized. Therefore, the stronger parallelization can be achieved in the scaled CVOCA via utilizing the proposed wavelength interleaving method.

We've added discussions accordingly as follows.

Supplementary Materials

Potentials of performance scaling

“Signal bandwidth. ... While the more closely spaced wavelengths are utilized for implementing multiple parallel computing channels, the Nyquist bandwidth of input signal can keep enough through utilizing wavelength interleaving method. Specifically, all used wavelengths with more closely spaced can be divided into 9 groups in sequence, the single parallel computing channel can be designed by selecting one wavelength at the same position from every group to make up the weights of one 3×3 kernel. As such, in the scaled CVOCA, if a microcomb with a smaller FSR is used, stronger parallelization can be achieved without sacrificing Nyquist bandwidth via utilizing the proposed wavelength interleaving technique.”

12. The numbering of references in the main text should be revised to reflect their order of appearance.

We appreciate the reviewer's comments, and have revised the numbering of references according to their order of appearance.